# INVERSE LINEAR BANDITS VIA LINEAR PROGRAMS

## ABSTRACT

Inverse reinforcement learning (IRL) is a well-established paradigm for circumventing the need for explicit reward. In this paper, we study the problem of estimating the reward function from a single sequence of actions (i.e., a demonstration) of a stochastic linear bandit algorithm. Our main result is a unified approach for inverse linear bandits, based on the idea of formulating a linear program by tightly characterizing the confidence intervals of pulled actions. We show that the estimation error of our algorithms matches the information-theoretic lower bound, up to polynomial factors in $d$ and $\log T$, where $d$ is the dimensionality of the feature space and $T$ is the length of the demonstration. Compared to prior approaches, our approach (i) gives a unified reward estimator that works when the demonstrator employs LinUCB or Phased Elimination, two popular algorithms for stochastic linear bandits, while existing estimator only works for Phased Elimination; (ii) does not require access to hyperparameters or internal states of the demonstrator algorithm as required by prior work; and (iii) works for general action sets, while existing estimator requires assumptions on the density and geometry of the action set. We further demonstrate the practicality of our new approach by validating our new algorithms on synthetic data and demonstrations constructed from real-world datasets, where our estimators significantly outperform existing ones.

## 1 INTRODUCTION

A central challenge in applying reinforcement learning (RL) to real-world problems is designing a suitable reward function. This process, often called reward engineering, is notoriously difficult and time-consuming (Anderson, 2001). More importantly, a misspecified reward can lead to undesirable agent behaviors, a phenomenon known as reward hacking (Amodei et al., 2016). Inverse Reinforcement Learning (IRL) (Ng & Russell, 2000; Abbeel & Ng, 2004) provides a powerful framework to address this challenge by formalizing the concept of learning from demonstration. Instead of manually crafting a reward function, the goal in IRL is to infer the underlying reward function that an expert demonstrator is implicitly optimizing. This paradigm has successfully powered applications in complex domains like robotics and autonomous driving (Abbeel et al., 2010; Ziebart et al., 2008).

Traditional IRL frameworks are built upon the assumption that the demonstrator's policy is optimal. While this provides a powerful learning signal, this stringent assumption introduces two critical challenges. First, it can lead to high sample complexity. When the demonstrator's policy is (near-)optimal, a single trajectory provides poor directional coverage, so recovering the underlying rewards typically requires multiple independent demonstrations. Second, and more fundamentally, it suffers from a well-known non-identifiability issue (Ng & Russell, 2000): a single policy can be optimal for an entire family of different reward functions. This ambiguity is a major obstacle because the recovered reward may not generalize correctly or capture the demonstrator's true underlying intent.

This work adopts a complementary observational regime: rather than assuming access to a *perfectly optimal policy*, we observe the actions of a *learning agent* as it converges to a (near-)optimal policy. This choice is motivated by two considerations. First, a demonstrator's learning trajectory is often more readily available than a static optimal policy; data from deployed systems like web services or robotic platforms naturally captures this learning process. Second, and more importantly, the sequence of actions taken by a learning algorithm is information-theoretically richer than a final deterministic optimal policy. The agent's exploration and suboptimal choices are not noise but

rather a crucial signal that directly reveals information about the relative quality of different actions, allowing us to break the non-identifiability deadlock and reduce sample complexity.

In fact, the above issues of sample complexity and identifiability occur even in the much simpler stochastic multi-armed bandit (MAB) setting. To resolve these issues, Guo et al. (2021) introduce the inverse bandit paradigm, where the goal is to estimate the reward structure by observing a *single* online demonstration of a low-regret algorithm. In particular, when the demonstrator employs Successive Arm Elimination (SAE) (Even-Dar et al., 2006) or Upper Confidence Bound (UCB) (Auer, 2002), Guo et al. (2021) show that the demonstrator's behavior (i.e., the sequence of arms picked by the demonstrator) would be sufficient to circumvent identifiability issues and the requirement of multiple demonstrations, and optimal reward estimators could be built based on that. A subsequent work (Guha et al., 2024) focused on the stochastic linear bandits setting and construct a reward estimator based on a single demonstration of the Phased Elimination algorithm, which serves as the first step towards understanding IRL in large action sets.

Meanwhile, the estimator by Guha et al. (2024) has the following limitations: (i) the estimator only works when the demonstrator employs Phased Elimination (Valko et al., 2014; Lattimore & Szepesvári, 2020); (ii) the estimator requires assumptions on the density and geometry of the action set; (iii) the estimator requires access to hyperparameters and internal states of the demonstrator algorithm. In fact, the first two limitations have been explicitly discussed in the work of Guha et al. (2024), and weakening these assumptions has been left as an open problem for future work.

In terms of methodology, the estimator by Guha et al. (2024) employs the least squares estimator, and only utilizes actions from the last epoch of the Phased Elimination demonstrator to form the final estimator. On the other hand, actions picked in any epoch of Phased Elimination reveal certain information about the ground truth reward parameter which could be utilized to obtain a more accurate estimate. E.g., if an action $a$ is picked in the $l$-th epoch, then it can be shown that the suboptimality gap of $a$ is at most $2^{-l}$. In this sense, using only actions from the last epoch is clearly insufficient for extracting all available information from the demonstration. This seems to be a fundamental limitation of the framework employed by Guha et al. (2024), as it is hard to utilize data with different accuracy levels in least squares estimate. This also explains why the estimator by Guha et al. (2024) works only for Phased Elimination, requires assumptions on the action set, and requires access to the internal states of the demonstrator.

In this paper, we show it is possible to address all the above limitations, using a unified approach, for the problem of inverse linear bandits. Our new idea is to formulate a linear program by tightly characterizing the confidence intervals of pulled actions to form the reward estimator. We show that the estimation error of our estimator matches the information-theoretic lower bound, up to polynomial factors in $d$ and $\log T$, where $d$ is the dimensionality of the feature space and $T$ is the length of the demonstration. Moreover, our approach (i) gives a unified reward estimator that works when the demonstrator employs LinUCB (Dani et al., 2008; Chu et al., 2011; Abbasi-Yadkori et al., 2011) or Phased Elimination (Valko et al., 2014; Lattimore & Szepesvári, 2020), two popular algorithms for stochastic linear bandits; (ii) does not require access to hyperparameters or internal states of the demonstrator algorithm; and (iii) works for general action sets. Besides the demonstration (the sequence of arms picked by the demonstrator), the only additional input required by our algorithm is (an approximation of) the best reward, which we show to be necessary for breaking symmetry. We further illustrate the practicality of our new approach by validating our new algorithms on synthetic data and demonstrations constructed from real-world datasets, where our estimators significantly outperform the estimator by Guha et al. (2024).

**Our Contributions.**    Below we give a more detailed description of our main contributions.

- We first show that when the demonstrator employs LinUCB or Phased Elimination, for any action $a$ in the action set, the estimation error of any reward estimator is lower bounded by a quantity related to $a$ and the inverse expected feature covariance matrix of the demonstrator. This lower bound serves as a concrete baseline when analyzing specific reward estimators.

- We develop a unified inverse estimator of the reward parameter from a single demonstration from the demonstrator, and the same estimator works when the demonstrator employs LinUCB or Phased Elimination. The estimation error of our estimator matches the aforementioned information-theoretic lower bound, up to polynomial factors in $d$ and $\log T$.

Our estimator requires only an approximation of the best reward, does not require access to hyperparameters or internal states of the demonstrator algorithm, and does not place any assumption on the density or geometry of the action set.

- Completing our theory, we perform experiments to validate our new estimators on synthetic data and demonstrations constructed from real-world datasets to illustrate the empirical superiority of our estimator over existing ones.

## 2 RELATED WORK.

**Stochastic Linear Bandits.**     Stochastic linear bandits were introduced by Abe & Long (1999). Since its introduction, various algorithms have been proposed (Auer, 2002; Dani et al., 2008; Chu et al., 2011; Abbasi-Yadkori et al., 2011; Agrawal & Goyal, 2013), and we refer readers to the monograph by Lattimore & Szepesvári (2020) for a comprehensive literature review. In this paper, we construct inverse reward estimators when the demonstrator employs LinUCB (Dani et al., 2008; Li et al., 2010; Chu et al., 2011; Abbasi-Yadkori et al., 2011) or Phased Elimination (Valko et al., 2014; Lattimore & Szepesvári, 2020), two popular algorithms for stochastic linear bandits. LinUCB and Phased Elimination could be regarded as generalizations of UCB (Auer, 2002) and SAE (Even-Dar et al., 2006), two demonstrators studied in the multi-armed bandits setting by Gao et al. (2018), to the linear bandits setting. Besides LinUCB and Phased Elimination, other algorithms (e.g., linear Thompson sampling (Agrawal & Goyal, 2013)) exist for linear bandits, and an interesting future direction is to generalize our techniques to handle more demonstrator algorithms.

**IRL.**     The framework of IRL is established by (Ng & Russell, 2000; Abbeel & Ng, 2004) to formalize the concept of learning from demonstration. However, non-identifiability is a well-known issue in IRL, and recent research on this issue (Cao et al., 2021; Lindner et al., 2022; Metelli et al., 2023) has emphasized the importance of incorporating more exploration in the expert's demonstrations. Nevertheless, existing works on IRL mostly focus on demonstrations generated by an optimal policy, though some works also consider sub-optimal experts (Gao et al., 2018; Brown et al., 2019; Poiani et al., 2024). Jacq et al. (2019) studied the problem of learning from a policy optimization algorithm, though they focus primarily on the empirical aspects.

As mentioned in the introduction, the inverse bandit paradigm was first introduced by Guo et al. (2021), where it has been shown that when the demonstrator employs SAE or UCB, optimal reward estimators could be built based on a single demonstration. This framework was later generalized to linear bandits by Guha et al. (2024). In this paper, we focus on the same inverse linear bandit setting as Guha et al. (2024), and our goal is to address the limitations of the estimators by Guha et al. (2024) mentioned earlier in the introduction.

## 3 PROBLEM FORMULATION

**Stochastic Linear Bandits.**     In the stochastic linear bandit setting, in each round $1 \le t \le T$, an algorithm selects an action $a_t$ from the action set $\mathcal{A} \subset \mathbb{R}^d$ and observes a reward $x_t = \langle a_t, \theta^* \rangle + \eta_t$. Here, $\theta^* \in \mathbb{R}^d$ is an unknown parameter with $\|\theta^*\|_2 \le \sqrt{d}$, the action set $\mathcal{A}$ satisfies $\|a\|_2 \le 1$ for all $a \in \mathcal{A}$, and the mean rewards are bounded with $\langle a, \theta^* \rangle \le 1$. The noise $\eta_t$ is conditionally 1-sub-Gaussian given existing observations. The goal is to minimize the pseudo-regret, defined as $R_T = \sum_{t=1}^{T} [\mu^* - \langle a_t, \theta^* \rangle]$, where $\mu^* = \max_{a \in \mathcal{A}} \langle a, \theta^* \rangle$ is the optimal reward value.

Below we introduce two stochastic linear bandits algorithms, which are the two algorithms the demonstrator might employ for the inverse linear bandits problem introduced later.

**Demonstrator I: Phased Elimination.**     Phased Elimination (Valko et al., 2014; Lattimore & Szepesvári, 2020) operates in sequential phases $\ell = 1, 2, ..., L$, maintaining an active arm set $\mathcal{A}_\ell$ that shrinks over time, where $L$ is the total number of phases. Each phase uses the G-optimal design (see Section A for the definition) to determine the allocation of pulls among remaining arms. This approach is known to achieve a regret bound of $O(d\sqrt{T \log T})$ (Lattimore & Szepesvári, 2020).

**Demonstrator II: LinUCB.** LinUCB (Dani et al., 2008) follows an optimistic approach by maintaining and updating a regularized least squares estimate of the reward parameter $\theta^*$. LinUCB is known to achieve a regret bound of $\mathcal{O}(d\sqrt{T}\log T)$ (Dani et al., 2008).

---

**Algorithm 1** Phased Elimination

**Require:** $\delta_{\mathsf{PE}}, T$
1: $\ell \leftarrow 0, \mathcal{A}_1 \leftarrow \mathcal{A}$
2: **while** Number of rounds $\leq T$ **do**
3:      $\varepsilon_\ell \leftarrow 2^{-\ell}$
4:      $\pi_\ell \leftarrow \text{G-OptimalDesign}(\mathcal{A}_\ell)$
5:      **for** each $a \in \mathcal{A}_\ell$ **do**
6:          $n_\ell(a) \leftarrow \frac{8d\pi_\ell(a)}{\epsilon_\ell^2}\left(d\log 6 + \log\left(\frac{\ell(\ell+1)}{\delta_{\mathsf{PE}}}\right)\right)$
7:      **end for**
8:      Play each $a \in \mathcal{A}_\ell$ exactly $n_\ell(a)$ times
9:      $V_\ell \leftarrow I_d + \sum_{a \in \mathcal{A}_\ell} n_\ell(a)aa^\top$
10:     $\hat{\theta}_\ell \leftarrow V_\ell^{-1}\sum_{t=t_\ell}^{t_\ell+n_\ell} a_t x_t$
11:     $\mathcal{A}_{\ell+1} \leftarrow \left\{a \in \mathcal{A}_\ell : \max_{b \in \mathcal{A}_\ell}\langle\hat{\theta}_\ell, b-a\rangle \leq 2\varepsilon_\ell\right\}$
12:     $\ell \leftarrow \ell+1$
13: **end while**

---

**Algorithm 2** LinUCB

**Require:** $\delta_{\mathsf{UCB}}, T$
1: $V_0 \leftarrow I_d, b_0 \leftarrow \mathbf{0}_d$
2: **for** $t = 1$ to $T$ **do**
3:      Observe $\mathcal{A}_t$
4:      $\hat{\theta}_{t-1} \leftarrow V_{t-1}^{-1}b_{t-1}$
5:      $\sqrt{\beta_t} \leftarrow \sqrt{d} + \sqrt{2\log\delta_{\mathsf{UCB}}^{-1} + d\log\left(\frac{d+t}{d}\right)}$
6:      $a_t \leftarrow \arg\max_{a \in \mathcal{A}}\left[a^\top\hat{\theta}_{t-1} + \sqrt{\beta_t}\|a\|_{V_{t-1}^{-1}}\right]$
7:      Play $a_t$, observe $x_t$
8:      $V_t \leftarrow V_{t-1} + a_t a_t^\top, b_t \leftarrow b_{t-1} + r_t a_t$
9: **end for**

---

**Inverse Linear Bandits.** We now formally define the inverse linear bandit problem. The inverse learner is assumed to have access only to the sequence of actions $(a_1, a_2, \ldots, a_T)$ produced by a single run of either Phased Elimination or LinUCB. Moreover, we have access to $\mu \in [\mu^*, \mu^* + \kappa]$, which is an approximation of the optimal reward value. Importantly, the inverse learner does *not* observe the corresponding rewards $(x_1, x_2, \ldots, x_T)$. Given these inputs, the inverse learner outputs an estimate $\hat{\theta}$ to recover the true reward parameter $\theta^*$ of the underlying linear bandit instance that generated the observed action sequence.

*Remark on the necessity of knowing $\mu$.* We prove in Appendix B that if we have no information about $\mu^*$, the demonstrations (sequence of actions) generated by Phased Elimination or LinUCB could be identical (i.e., follow the same distribution) for different values of $\mu^*$. Therefore, it is necessary to know $\mu^*$ (or at least approximately) to break symmetry and to enable learning the reward parameter $\theta^*$. Moreover, similar to prior work (Guo et al., 2021; Guha et al., 2024), when $\mu^*$ is unknown, we may restrict ourselves to estimating rewards up to additive shift of $\mu^*$.

## 4   FUNDAMENTAL LIMITS ON INVERSE REWARD ESTIMATION

In this section, we present an information-theoretic lower bound that applies to any inverse learner. Our lower bound shows that, for a fixed action $a \in \mathcal{A}$, the estimation error of any estimator is lower bounded by $\Omega(\|a\|_{\bar{V}^{-1}})$. Here, $\bar{V}$ is the expected feature covariance matrix of the demonstrator. This lower bound will later serve as a concrete baseline when analyzing specific reward estimators.

**Theorem 1.** *Consider a $T \geq 3$-round linear bandit $\mathcal{M}$ with standard Gaussian noise, where $\theta \in \mathbb{R}^d$ is the true reward parameter and $\mathcal{A}$ is the action set. For any demonstrator algorithm, for any action $a \in \mathcal{A}$, there exists another linear bandit instance $\mathcal{M}'_a$ with parameter $\theta'_a \in \mathbb{R}^d$ and the same action set $\mathcal{A}$, such that for any estimator $\Psi$ that maps the sequence of actions $(a_1, \ldots, a_T)$ to a parameter estimate $\hat{\theta} = \Psi(a_1, \ldots, a_T)$,*

$$\max_{\nu \in \{\theta, \theta'_a\}} \mathbb{P}_\nu\left(|a^\top(\hat{\theta} - \nu)| \geq \frac{1}{2}\|a\|_{\bar{V}^{-1}}\right) \geq \frac{1}{12},$$

*where $\mathbb{P}_\nu$ denotes the probability distribution over action sequences induced by the demonstrator under a linear bandit instance with parameter $\nu$ and $\overline{V} = I + \mathbb{E}_\theta\left[\sum_{t=1}^T a_t a_t^\top\right]$ is the expected feature covariance matrix of the demonstrator under instance $\mathcal{M}$.*

*Moreover, when the demonstrator algorithm employs Phased Elimination or LinUCB with $\delta_{\mathsf{PE}} \leq 1/T$ or $\delta_{\mathsf{UCB}} \leq 1/T$, the above result holds even if the estimator has access to $\mu$ which is an approximation of the optimal reward value satisfying $\mu \in [\mu^*, \mu^* + \kappa]$ with $\kappa = O\left(d \log T / \sqrt{T}\right)$.*

Our proof is based on the change of distribution argument (Kaufmann et al., 2016). In our formal proof, we actually prove a stronger statement: even if the estimator has access to the sequence of rewards observed by the demonstrator, the same lower bound still holds. Note that providing more information to the estimator only makes the lower bound *stronger*. Meanwhile, all our estimators *do not* rely on access to the sequence of rewards observed by the demonstrator, as stated in Section 3.

Intuitively, $\|a\|_{\overline{V}^{-1}}$ quantifies the amount of information the demonstrator has provided for a specific action $a$. When specializing to the special case of multi-armed bandits, the lower bound in Theorem 1 is equivalent to the hardness result in Guo et al. (2021) (which works only for multi-armed bandits). To see this, we may associate each arm in multi-armed bandits with a standard basis vector, in which case our lower bound is equivalent to $\min\{\frac{1}{\mathbb{E}[\sqrt{n_i}]}, 1\}$, where $n_i$ is the number of times that the demonstrator algorithm pulls action $i$. Therefore, Theorem 1 could be seen as a generalization of the hardness result in Guo et al. (2021) to the linear bandits setting. Indeed, Theorem 1 is also proved using the change of distribution argument Kaufmann et al. (2016), though a different argument is needed to relate the amount of uncertainty to $\|a\|_{\overline{V}^{-1}}$.

# 5 WARMUP: REWARD ESTIMATORS FOR PHASED ELIMINATION

In this section, we present reward estimators for Phased Elimination (Algorithm 3). These estimators serve as a warmup for the more complicated estimators in Section 6.

The high-level idea behind our estimator is simple: when the demonstrator algorithm is Phased Elimination, if an action $a_t$ is pulled in the $\ell$-th phase, by standard analysis, we are certain that $a_t^\top \theta^* \in [\mu^* - 8 \cdot 2^{-\ell}, \mu^*]$, since otherwise $a$ would have already been eliminated in previous phases. Assuming access to $\mu \in [\mu^*, \mu^* + \kappa]$, we then know that $a_t^\top \theta^* \in [\mu - 8 \cdot 2^{-\ell} - \kappa, \mu]$, which we add into the set of constraints. The final estimate $\hat{\theta}$, is simply an arbitrary vector satisfying all the above constraints. Note that all constraints we have imposed are linear constraints w.r.t. the unknown parameters, which means the final program is a linear program (LP) which can be efficiently solved.

---

**Algorithm 3** Reward Estimator for Phased Elimination

1: **Input:** sequence of actions executed by the demonstrator $(a_1, a_2, \ldots, a_T)$, $\mu \in [\mu^*, \mu^* + \kappa]$
2: Set $\hat{\theta}$ to be a vector satisfying

$$\forall a_t, \mu - 8 \cdot 2^{-\ell_t} - \kappa \leq \hat{\theta}^\top a_t \leq \mu,$$

where $\ell_t \geq 1$ is the phase that the action $a_t$ is pulled.
3: **return** $\hat{\theta}$

---

**Analysis of Algorithm 3.** The following theorem characterizes the performance of Algorithm 3.

**Theorem 2.** *With probability at least $1 - \delta_{\mathsf{PE}}$, the LP in Algorithm 3 has a feasible solution. Moreover, let $\hat{\theta}$ be a feasible solution of the LP in Algorithm 3, for any action $a \in \mathcal{A}$, $|a^\top(\hat{\theta} - \theta^*)| \leq O\left(\sqrt{d^2 \log T \log(L/\delta_{\mathsf{PE}})}\right) \cdot \|a\|_{\hat{V}^{-1}} + \sqrt{d}\kappa$, where $\hat{V} = \sum_{t=1}^T a_t a_t^\top + I$.*

Theorem 2 shows that for a fixed action $a$, the estimation error of Algorithm 3 is upper bounded by $O\left(\sqrt{d^2 \log T \log(L/\delta_{\mathsf{PE}})}\right) \cdot \|a\|_{\hat{V}^{-1}} + \sqrt{d}\kappa$, where $\hat{V}$ is the empirical feature covariance matrix of the demonstrator. On the other hand, the lower bound in Theorem 1 shows that the estimation error is lower bounded by $\|a\|_{\overline{V}^{-1}}$, where $\overline{V}$ is the expected feature covariance matrix of the demonstrator. At this point, a natural question is whether the empirical feature covariance matrix $\hat{V}$ will concentrate around its mean $\overline{V}$. Note that this is a question purely related to the demonstrator algorithm, and has nothing to do with the inverse estimator. In the following theorem, we show that at least for Phased Elimination, it is indeed the case.

**Theorem 3.** *In Phased Elimination, if the failure probability $\delta_{\mathsf{PE}}$ satisfies $d \log 6 + \log \left( \frac{L(L+1)}{\delta_{\mathsf{PE}}} \right) \geq$*

*$(1+c) \left( d \log 6 + \log \left( \frac{L(L+1)}{1/T} \right) \right)$ for any $c > 0$, then $\overline{V} \preceq O(d^2) \cdot \hat{V}$ holds with probability $1 - 1/T$.*

*Remark on the constraint on $\delta_{\mathsf{PE}}$.* To show that the empirical feature covariance matrix $\hat{V}$ of Phased Elimination concentrates around its mean, Theorem 3 put a constraint on $\delta_{\mathsf{PE}}$, which, roughly speaking, requires the algorithm to use a confidence interval with length slightly longer than that when $\delta_{\mathsf{PE}} = 1/T$. First, such constraint is on the parameter used by the demonstrator algorithm when generating the demonstration, and our inverse reward estimator *does not require access to $\delta_{\mathsf{PE}}$*. Moreover, such constraint on $\delta_{\mathsf{PE}}$ is necessary even in the multi-armed bandit case. To see this, in any phase of the algorithm, for any $a \in \mathcal{A}$, if the upper confidence bound of $a$ is equal to its ground truth mean reward, while the lower confidence bound of $a^*$ is equal to $\mu^*$, then $a$ would be eliminated regardless of the suboptimality gap of $a$, in which case the concentration result fails. By slightly increasing the confidence interval, we make sure that Phased Elimination maintains sufficient exploration across all relevant directions in the action set. On the other hand, such requirement only increases the regret bound by a small constant factor. We also note that in practice, exact confidence interval calculation is rarely feasible, and in most cases upper bounds are used, in which case our constraint automatically holds. Moreover, even if the constraint on $\delta_{\mathsf{PE}}$ does not hold, Theorem 2 still holds, albeit the estimation error might not match the information-theoretic lower bound in Theorem 1 in this case. Finally, we remark that such constraint was also implicitly used in prior work (Guo et al., 2021; Guha et al., 2024), and we choose to make it explicit for transparency.

Combining Theorem 2 with Theorem 3, it is clear that for any action $a$, the estimation error of Algorithm 3 is upper bounded by $O \left( \sqrt{d^4 \log T \log(L/\delta_{\mathsf{PE}})} \right) \cdot \|a\|_{\overline{V}^{-1}} + \sqrt{d}\kappa$, where $\|a\|_{\overline{V}^{-1}}$ is the best estimation error of *any possible* reward estimator. Therefore, Algorithm 3 is no worse than any other estimator up to a factor of $O \left( \sqrt{d^4 \log T \log(L/\delta_{\mathsf{PE}})} \right)$ and an additional term of $\sqrt{d}\kappa$.

**Implementing Algorithm 3.** As mentioned earlier, the constraints in Algorithm 3 form an LP with $d$ variables. Moreover, since there are at most $O(d^2)$ unique actions pulled at each phase $\ell$, the total number of constraints is $O(d^2 L)$ where $L$ is the number of phases of Phased Elimination. Therefore, to implement Algorithm 3, the only non-trivial part is to calculate $\ell_t$, which is the phase that the action $a_t$ is pulled in Phased Elimination. $\ell_t$ could be exactly calculated in many scenarios. For instance, if we have access to certain hyperparameters (e.g. the failure probability $\delta_{\mathsf{PE}}$) of the demonstrator (Algorithm 1), we would know the total number of pulls in each phase, in which case $\ell_t$ could be easily calculated. Moreover, if the G-optimal design of Algorithm 1 has exactly $\frac{d(d+1)}{2}$ actions on its support in each phase $\ell$, then we could simply set $\ell_t = \ell_{t-1} + 1$ after encountering $\frac{d(d+1)}{2}$ unique actions on the sequence of actions executed by the demonstrator $(a_1, a_2, \ldots, a_T)$, and set $\ell_t = \ell_{t-1}$ otherwise. Below we give a better approach (Algorithm 4), which does not rely on access to hyperparameters or implementation details of the demonstrator.

---

**Algorithm 4** Reward Estimator for Phased Elimination

1: **Input:** sequence of actions executed by the demonstrator $(a_1, a_2, \ldots, a_T)$, $\mu \in [\mu^*, \mu^* + \kappa]$
2: Set $C \geq 0$ to be the smallest real number, so that the following constraints are feasible:

$$\forall a_t, \mu - C \cdot \sqrt{d^2/t} - \kappa \leq \hat{\theta}^\top a_t \leq \mu.$$

3: **return** a feasible solution $\hat{\theta}$ of the above constraints for the $C$ found in Step 2.

---

In Algorithm 4, for each action $a_t$, we add a constraint

$$\mu - C \cdot \sqrt{d^2/t} - \kappa \leq \hat{\theta}^\top a_t \leq \mu,$$

where $C$ is a global parameter. Our approach finds the smallest $C \geq 0$ so that the LP is still feasible. In order to find the smallest $C$, we could use a binary search. Specifically, each time we pick the middle point of the current interval as a candidate choice of $C$. If the LP is feasible under the current choice of $C$, we discard all choices of $C$ smaller than our current choice. Otherwise, all choices of

*C* larger than our current choice would be discarded. Repeating this process allows us to identify the smallest feasible *C* up to the desired precision.

The intuition behind this approach, is that for each action $a_t$, the phase $\ell$ that $a_t$ is pulled satisfies $2^{-\ell} = O(\sqrt{d^2 \log(\ell/\delta_{\mathsf{PE}})/T})$. Therefore, by taking $C = O(\sqrt{\log(L/\delta_{\mathsf{PE}})})$, the LP must be feasible, in which case Algorithm 4 recovers the constraints in Algorithm 3 (up to constants in the constraints), which implies the estimation error of Algorithm 4 is close to that of Algorithm 3. The following theorem formalizes the above intuition.

**Theorem 4.** *Let $\hat{\theta}$ be a feasible solution of the LP in Algorithm 4. With probability at least $1 - \delta_{\mathsf{PE}}$, for any action $a \in \mathcal{A}$, $|a^\top(\hat{\theta} - \theta^*)| \leq O\left(\sqrt{d^2 \log T \log\left((L/\delta_{\mathsf{PE}})\right)} \cdot \|a\|_{\hat{V}^{-1}} + \sqrt{d}\kappa\right)$, where $\hat{V} = \sum_{t=1}^T a_t a_t^\top + I$.*

Note that by minimizing $C$, we ensure that the constraints are as tight as possible, which could potentially lead to a smaller estimation error. Moreover, for duplicate actions $a_t$ in the sequence, we can keep only the tightest constraint (i.e., the one with the largest lower bound) for each distinct action. Since the number of distinct actions is at most $O(d^2 L)$ in Phased Elimination, the number of constraints is effectively reduced to $O(d^2 L)$, ensuring computational efficiency.

Compared to Algorithm 3, the main advantage of Algorithm 4 is that it is easy to implement. Clearly, Algorithm 4 does not require hyperparameters or implementation details of the demonstrator algorithm. Moreover, Algorithm 4 adaptively finds the tightest constraints so that the LP is still feasible. This avoids defining exact constants in the linear program (as in Algorithm 3). In Section 7, we further demonstrate that such adaptive constraints lead to better practical performance.

# 6 UNIFIED INVERSE ESTIMATOR FOR LINUCB AND PHASED ELIMINATION

## 6.1 REWARD ESTIMATOR FOR LINUCB

In this section, we present reward estimators for LinUCB. The construction follows a similar philosophy to Phased Elimination but adapts to the case when applied to LinUCB.

The key insight is that when LinUCB selects an action $a_t$ at time $t$, its UCB value $a_t^\top \hat{\theta}_{t-1} + \beta_t \|a_t\|_{V_{t-1}^{-1}}$ must be at least that of the optimal action. Since the optimal action's expected reward is $\mu^*$, and we have access to $\mu \in [\mu^*, \mu^* + \kappa]$, we can derive that $a_t^\top \theta^*$ must satisfy:

$$a_t^\top \theta^* \geq \mu - 2\beta_t \|a_t\|_{V_{t-1}^{-1}} - \kappa.$$

This lower bound comes from considering the estimation error of both the selected action and the optimal action, while the upper bound $a_t^\top \theta^* \leq \mu$ follows directly from the definition of $\mu^*$.

---

**Algorithm 5** Reward Estimator for LinUCB

---

1: **Input:** sequence of actions executed by the demonstrator $(a_1, a_2, \ldots, a_T)$, $\mu \in [\mu^*, \mu^* + \kappa]$
2: Set $C \geq 0$ to be the smallest real number, so that the following constraints are feasible:

$$\forall a_t, \mu - C \cdot \|a\|_{V_{t-1}^{-1}} - \kappa \leq \hat{\theta}^\top a_t \leq \mu.$$

3: **return** a feasible solution $\hat{\theta}$ of the above constraints for the $C$ found in Step 2.

---

**Analysis of Algorithm 5.** The following theorem provides a performance guarantee.

**Theorem 5.** *Let $\hat{\theta}$ be a feasible solution of the LP in Algorithm 4. With probability at least $1 - \delta_{\mathsf{UCB}}$, for any action $a \in \mathcal{A}$, $|a^\top(\hat{\theta} - \theta^*)| \leq O\left(\sqrt{d^2 \log^2 T \log \delta_{\mathsf{UCB}}^{-1}} \cdot \|a\|_{\hat{V}^{-1}} + \sqrt{d}\kappa\right)$, where $\hat{V} = \sum_{t=1}^T a_t a_t^\top + I$.*

Recall that the lower bound in Theorem 1 shows that the estimation error is lower bounded by $\|a\|_{\overline{V}^{-1}}$, where $\overline{V}$ is the expected feature covariance matrix of the demonstrator. Similar to the Phased Elimination case, for LinUCB, we may ask whether the empirical feature covariance matrix $\hat{V}$ will concentrate around its mean $\overline{V}$. The following theorem shows that similar concentration results can indeed be established for LinUCB.

**Theorem 6.** *For LinUCB, if $\delta_{\sf UCB}$ satisfies $\sqrt{d} + \sqrt{2\log \delta_{\sf UCB}^{-1} + d\log(1 + T/d)} \geq (1 + c)\left(\sqrt{d} + \sqrt{2\log T + d\log(1 + T/d)}\right)$ for any $c > 0$, then $\overline{V} \preceq O(d^5 \log^5 T \log \delta_{\sf UCB}^{-1})\hat{V}$ holds with probability $1 - 1/T$.*

*Remark on Theorem 6 and the constraint on $\delta_{\sf UCB}$.* Here we also put a constraint on $\delta_{\sf UCB}$ for the same reason as the Phased Elimination case. We remark that such constraint was also implicitly used in the multi-armed bandit case in prior work (Guo et al., 2021), and our inverse learner does not require access to $\delta_{\sf UCB}$. We also remark that the proof of Theorem 6 is highly non-trivial and is the most technically involved part of the paper. The proof requires structural characterizations of LinUCB by comparing its behavior with Phased Elimination, to prove the final concentration result, which further implies the optimality of our approach (up to polynomial factors in $d$ and $\log T$).

Combining Theorems 5 and 6, we obtain that the estimation error of Algorithm 5 is bounded by $O\left(\sqrt{d^7 \log^7 T \log^2 \delta_{\sf UCB}^{-1}}\right) \cdot \|a\|_{\overline{V}^{-1}} + \sqrt{d}\kappa$. This establishes that the estimation error of our estimator matches the information-theoretic lower bound up to a factor of $O\left(\sqrt{d^7 \log^7 T \log^2 \delta_{\sf UCB}^{-1}}\right)$ and an additional term of $\sqrt{d}\kappa$.

## 6.2 Unified Reward Estimator

In this section, we present a unified reward estimation framework that simultaneously accommodates both Phased Elimination and LinUCB. This approach addresses the challenge of designing a unified estimator that remains feasible regardless of which algorithm the demonstrator employs.

The key insight is to construct constraints that capture the essential properties of both algorithms. For Phased Elimination, the estimation error typically scales as $O(\sqrt{d^2/t})$ due to its phased structure and deterministic elimination process. For LinUCB, the error bound involves $O(\sqrt{d}\|a_t\|_{V_{t-1}^{-1}})$ stemming from its confidence-bound based exploration. By taking the maximum of these two quantities, we ensure that the constraints contain the true parameter $\theta^*$ with high probability, regardless of the demonstrator algorithm used to generate the demonstration.

---

**Algorithm 6** Unified Reward Estimator for Phased Elimination and LinUCB

1: **Input:** sequence of actions executed by the demonstrator $(a_1, a_2, \ldots, a_T)$, $\mu \in [\mu^*, \mu^* + \kappa]$
2: Set $C \geq 0$ to be the smallest real number such that the following constraints are feasible:

$$\forall t \in [T], \quad \mu - C \cdot \max\left(\sqrt{d}\|a_t\|_{V_{t-1}^{-1}}, \sqrt{d^2/t}\right) - \kappa \leq \hat{\theta}^\top a_t \leq \mu.$$

3: **return** a feasible solution $\hat{\theta}$ of the above constraints for the $C$ found in Step 2.

---

The following theorem establishes the theoretical guarantees for our unified estimator.

**Theorem 7.** *For any demonstration sequence generated by either Phased Elimination or LinUCB, for any feasible solution $\hat{\theta}$ of Algorithm 6 and any action $a \in \mathcal{A}$,*

1. *For Phased Elimination, with probability at least $1 - \delta_{\sf PE}$,*

$$|a^\top (\hat{\theta} - \theta^*)| \leq O\left(\sqrt{d^2 \log T \log\left(L\delta_{\sf PE}^{-1}\right)}\right) \cdot \|a\|_{\hat{V}^{-1}} + \sqrt{d}\kappa,$$

2. *For LinUCB, with probability at least $1 - \delta_{\sf UCB}$,*

$$|a^\top(\hat{\theta} - \theta^*)| \le O\left(\sqrt{d^2 \log^2 T \log \delta_{\mathsf{UCB}}^{-1}}\right) \cdot \|a\|_{\hat{V}^{-1}} + \sqrt{d}\kappa,$$

*where* $\hat{V} = \sum_{t=1}^{T} a_t a_t^\top + I$.

The unified estimator offers several advantages. First, it provides a single framework that works seamlessly with both Phased Elimination and LinUCB, eliminating the need for algorithm-specific estimators. Second, the estimation error gracefully adapts to the characteristics of the underlying demonstrator algorithm—achieving the $O\left(\sqrt{d^2 \log^2 T \log \delta_{\mathsf{UCB}}^{-1}}\right) \cdot \|a\|_{\hat{V}^{-1}} + \sqrt{d}\kappa$ bound for Lin-UCB and the $O\left(\sqrt{d^2 \log T \log\left(L\delta_{\mathsf{PE}}^{-1}\right)}\right) \cdot \|a\|_{\hat{V}^{-1}} + \sqrt{d}\kappa$ bound for Phased Elimination. Our approach demonstrates a carefully designed estimator can effectively learn from demonstrations without prior knowledge of the specific algorithm employed by the demonstrator.

## 7 EXPERIMENTS

The objective of our experiments is to evaluate the performance of our proposed phased elimination algorithms in comparison to the baseline algorithm by Guha et al. (2024). Since the baseline are based on the phased elimination framework, our evaluation exclusively focuses on this class of algorithms. Our evaluation consists of two parts: simulation experiments with synthetic data and semi-synthetic experiments. For all experiments, we set the maximum number of phases $L = 8$.

To validate the correctness of our statement at the end of Section 5, we first compare the standard version of Algorithm 3 with its binary search version.
As shown in Figure 1, the binary search version significantly outperforms the standard version. Therefore, throughout the experiments, we use the binary search version of Algorithm 3 for all evaluations.

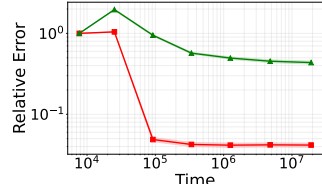

Figure 1: $d = 8, ord = 2$. (red: binary search version, green: standard version)

**Simulation Experiments.** In the simulated environments, we construct action sets by sampling 4000 vectors from the surface of the unit $\ell_1$, $\ell_2$, and $\ell_5$ balls. We run 100 trials of bandit instances with dimensions $d \in \{8, 16\}$. For each trial, we execute three algorithms: the baseline Algorithm Guha et al. (2024), our Algorithm 3 (binary search version), and our Algorithm 4. The relative error of the estimated $\hat{\theta}$ is computed as $\|\hat{\theta} - \theta^*\|_2/\|\theta^*\|_2$ at the final round of the last phase.

The results are visualized in log-log plots (Figure 2), which show the relative error as a function of the total number of rounds $T$.

**Semi-Synthetic Experiments.** To assess performance in more realistic scenarios, we use the Movie-Lens 25M dataset Lam & Herlocker (2006); Harper & Konstan (2015) and the Amazon Reviews digital music subset Hou et al. (2024). We repeat this process for 100 randomly selected users and average the relative error of $\hat{\theta}$. The results are plotted in Figure 2g-h.

Crucially, we observe that both Algorithm 3 (binary search version) and Algorithm 4 consistently outperform Algorithm Guha et al. (2024) across all scenarios.

## 8 CONCLUSION

We present a unified framework for inverse linear bandits that accurately estimates reward functions from single demonstrations, addressing key limitations of prior estimators. Future work includes extending our approach to other linear bandit algorithms like Thompson sampling and nonlinear settings, as well as tightening the $d$ and $\log T$ factors.

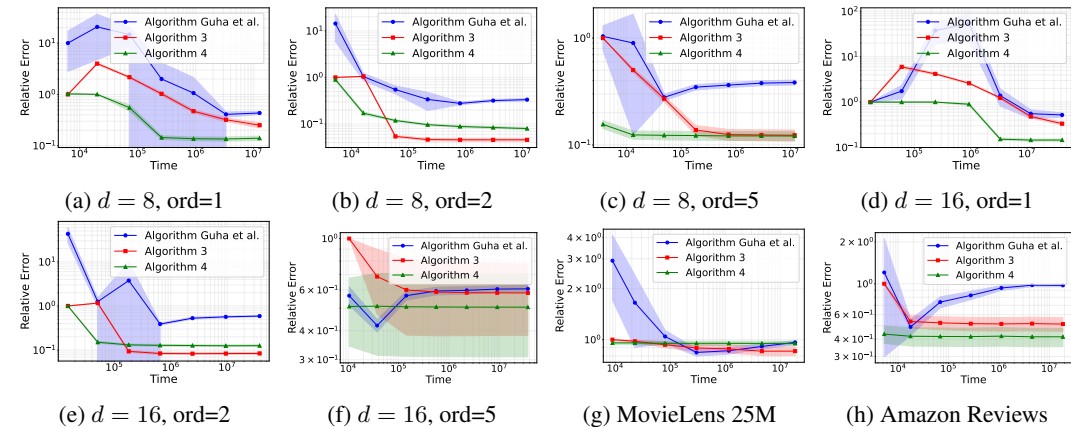

(a) $d = 8$, ord=1     (b) $d = 8$, ord=2     (c) $d = 8$, ord=5     (d) $d = 16$, ord=1

(e) $d = 16$, ord=2     (f) $d = 16$, ord=5     (g) MovieLens 25M     (h) Amazon Reviews

Figure 2: Experimental results: (a)-(d) $d = 8$ and $d = 16$ simulations; (e)-(h) additional simulations and semi-synthetic datasets. Our algorithms consistently outperform the baseline.

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

## THE USE OF LARGE LANGUAGE MODELS

In the preparation of this paper, we employed large language models (LLMs) as a general-purpose assistive tool for non-substantive tasks. Specifically, LLMs were used for:

- **Text-related work**: Assisting in editing, proofreading, and formatting of textual content.
- **Generation of plotting code**: Generating code snippets for creating figures and visualizations, which were then verified and customized by the authors.
- **Paper querying**: Facilitating literature searches and information retrieval to support background research.

The LLMs did not contribute to the core ideation, analysis, or substantive writing of the research. The authors take full responsibility for the entire content of this paper, including any portions influenced by LLMs, and affirm that the use of LLMs complies with academic integrity standards. LLMs are not considered authors or contributors to this work.

## A    PRELIMINARY LEMMAS AND DEFINITIONS

**Definition 1** (G-optimal design). *A G-optimal design for an action set $\mathcal{A}$ is a function $\pi : \mathcal{A} \to [0,1]$ that minimises $g(\pi) = \max_{a \in \mathcal{A}} \|a\|^2_{V(\pi)^{-1}}$ subject to $\sum_{a \in \mathcal{A}} \pi(a) = 1$, where $V(\pi) = \sum_{a \in \mathcal{A}} \pi(a) a a^\top$.*

**Lemma 1.** *For sets $A \subseteq B \subseteq \mathbb{R}^d$, let $\pi_A$ be any distribution over $A$, and let $\pi_B$ be G-optimal designs over $B$. Then:*

$$V_A(\pi_A) \preceq d \cdot V_B(\pi_B)$$

*Proof of Lemma 1.* By definition of G-optimal design:

$$\max_{b \in B} b^\top V_B(\pi_B)^{-1} b \leq d$$

Since $A \subseteq B$, for all $a \in A$:

$$a^\top V_B(\pi_B)^{-1} a \leq d$$

Define the matrix:

$$M = V_B(\pi_B)^{-1/2} V_A(\pi_A) V_B(\pi_B)^{-1/2}$$

Compute its trace:

$$\text{trace}(M) = \text{trace}\left(V_A(\pi_A) V_B(\pi_B)^{-1}\right)$$

$$= \text{trace}\left(\sum_{a \in A} \pi_A(a) a a^\top V_B(\pi_B)^{-1}\right)$$

$$= \sum_{a \in A} \pi_A(a) \text{trace}\left(a a^\top V_B(\pi_B)^{-1}\right)$$

Since $\text{trace}(a a^\top V_B(\pi_B)^{-1}) = a^\top V_B(\pi_B)^{-1} a$:

$$\text{trace}(M) = \sum_{a \in A} \pi_A(a) a^\top V_B(\pi_B)^{-1} a \leq \sum_{a \in A} \pi_A(a) \cdot d = d$$

The maximum eigenvalue satisfies:

$$\lambda_{\max}(M) \leq \sum \lambda_i(M) = \text{trace}(M) \leq d$$

Thus $M \preceq d \cdot I$, meaning:

$$z^\top M z \leq d \cdot z^\top z, \quad \forall z \in \mathbb{R}^d$$

Substituting $M$:

$$z^\top V_B(\pi_B)^{-1/2} V_A(\pi_A) V_B(\pi_B)^{-1/2} z \leq d \cdot z^\top z, \quad \forall z$$

Make the substitution $y = V_B(\pi_B)^{-1/2} z$ (so $z = V_B(\pi_B)^{1/2} y$):

$$y^\top V_A(\pi_A) y \leq d \cdot (V_B(\pi_B)^{1/2} y)^\top (V_B(\pi_B)^{1/2} y) = d \cdot y^\top V_B(\pi_B) y, \quad \forall y \in \mathbb{R}^d$$

Therefore:

$$V_A(\pi_A) \preceq d \cdot V_B(\pi_B)$$

$\square$

**Approximate G-optimal design** For our analysis in LinUCB part, we only require the existence of a finite-support $\varepsilon$-approximation of the G-optimal design. Formally, for any action set $\mathcal{A} \subseteq \mathbb{R}^d$ and any $\varepsilon \in (0,1)$, there exists a weighted subset $\widetilde{\mathcal{S}} \subseteq \mathcal{A}$ with weights $\{\tilde{w}_a\}_{a \in \widetilde{\mathcal{S}}}$ such that

$$(1-\varepsilon)V(\pi^*) \preceq \sum_{a \in \widetilde{\mathcal{S}}} \tilde{w}_a \, aa^\top \preceq (1+\varepsilon)V(\pi^*),$$

where $V(\pi^*)$ denotes the moment matrix of the exact G-optimal design. Moreover, such an $\varepsilon$-approximate G-optimal design can be chosen with support size

$$|\widetilde{\mathcal{S}}| = \mathcal{O}\left(\frac{d}{\varepsilon^2}\right),$$

as guaranteed by the spectral sparsification result of Batson–Spielman–Srivastava (2012). No explicit construction is required for our algorithm; we only use the fact that such a design exists.

**Lemma 2.** *The following bounds hold for the Phased-elimination algorithm:*

1. *$t \leq C\, d^2 \log(l_t/\delta) \cdot 4^{l_t}$ for all $l_t \in [2, L]$*

2. *$t \geq c\, d \log(l_t/\delta) \cdot 4^{l_t}$ for all $l_t \in [2, L]$*

3. *For all $\ell \in [2, L]$, for all $a \in \mathcal{A}_\ell$, we have $a^T \hat{\theta}_{\ell-1} \in [a^T \theta^* - \epsilon_{\ell-1}, a^T \theta^* + \epsilon_{\ell-1}]$ Lattimore & Szepesvári (2020)*

*where $c$ and $C$ are absolute constants.*

*Proof.* We prove the two bounds separately. Let $n_\ell$ denote the total number of rounds in phase $\ell$ of the algorithm, i.e., $n_\ell = \sum_{a \in \mathcal{A}_\ell} n_\ell(a)$. Note that phase indices start from 1, but for $l_t \in [2, L]$, we consider phases $\ell \geq 2$.

**Proof of (1): Upper bound.** From the algorithm, the total number of rounds in phase $\ell$ is:

$$n_\ell = \sum_{a \in \mathcal{A}_\ell} n_\ell(a) = \frac{8d}{\epsilon_\ell^2}\left(d \log 6 + \log\left(\frac{\ell(\ell+1)}{\delta}\right)\right),$$

where $\epsilon_\ell = 2^{-\ell}$, so $\epsilon_\ell^2 = 4^{-\ell}$, and thus:

$$n_\ell = 8d \cdot 4^\ell \left(d \log 6 + \log\left(\frac{\ell(\ell+1)}{\delta}\right)\right).$$

The cumulative time up to phase $\ell$ is $T_\ell = \sum_{k=1}^\ell n_k$. For $\ell \geq 2$, we have:

$$T_\ell = n_1 + \sum_{k=2}^\ell n_k.$$

Note that $n_1$ is a constant. For $k \geq 2$, we bound $n_k$:

$$n_k \leq 8d \cdot 4^k \left(d \log 6 + \log\left(\frac{k^2}{\delta}\right)\right) = 8d \cdot 4^k \left(d \log 6 + 2\log k + \log(1/\delta)\right).$$

Using geometric series properties:

$$\sum_{k=2}^\ell 4^k \leq \frac{4}{3} 4^\ell, \quad \sum_{k=2}^\ell 4^k \log k \leq \frac{4}{3} 4^\ell \log \ell.$$

Thus:

$$\sum_{k=2}^\ell n_k \leq 8d \left(d \log 6 + \log(1/\delta)\right) \cdot \frac{4}{3} 4^\ell + 16d \cdot \frac{4}{3} 4^\ell \log \ell.$$

Combining:

$$T_\ell \leq n_1 + \frac{32}{3} d \left(d \log 6 + \log(1/\delta)\right) 4^\ell + \frac{64}{3} d 4^\ell \log \ell.$$

For $d \geq 1$ and $\delta \leq 1$, we have $d \log 6 + \log(1/\delta) \leq d^2 \log(6/\delta)$ and $d \log \ell \leq d^2 \log \ell$, so there exists an absolute constant $C_1$ such that:

$$T_\ell \leq C_1 d^2 \log(\ell/\delta) 4^\ell.$$

Let $l_t$ be the phase at time $t$, so $t \leq T_{l_t}$. Therefore:

$$t \leq T_{l_t} \leq C_1 d^2 \log(l_t/\delta) 4^{l_t},$$

which proves (1) with $C = C_1$.

**Proof of (2): Lower bound.** For phase $\ell \geq 2$, we have:

$$n_\ell \geq 8d^2 \cdot 4^\ell,$$

Taking $c = 8$ immediately we hve:

$$t \geq cd^2 \cdot 4^{\ell_t}$$

which proves (2). $\qquad\qquad\qquad\qquad\qquad\qquad\qquad\qquad\qquad\qquad\qquad\qquad\qquad$ $\square$

**Lemma 3.** *Lattimore & Szepesvári (2020)The following bounds hold for the LinUCB algorithm:*

1. *The regret bound: $\hat{R}_T \leq C_R d\sqrt{T} \log T$*

2. *The confidence parameter: $\sqrt{\beta_t} = \sqrt{d} + \sqrt{2 \log(1/\delta) + d \log\left(\dfrac{d+t}{d}\right)}$*

3. *The sum of squared norms: $\sum_{t=1}^{T} \|A_t\|_{V_{t-1}^{-1}}^2 \leq 2d \log T$*

## B    PROOF OF REMARK ON THE NECESSITY OF KNOWING $\mu^*$

Consider the scenario where all actions $a \in \mathcal{A}$ have their last dimension equal to 1, i.e., $a = [a_1, a_2, \ldots, a_{d-1}, 1]^\top$. Let $\theta^* = [\theta_1^*, \theta_2^*, \ldots, \theta_{d-1}^*, \theta_d^*]^\top$ be the true parameter.

We aim to prove that for any two parameters $\theta^*$ and $\theta'^*$ such that $\theta_i^* = \theta_i'^*$ for $i = 1, \ldots, d-1$ but $\theta_d^* \neq \theta_d'^*$, the distribution of action sequences $a_1, \ldots, a_T$ generated by either the Phased Elimination algorithm (Algorithm 1) or the LinUCB algorithm (Algorithm 2) is identical. This implies that the algorithms' behavior is insensitive to the last dimension of $\theta^*$, making it impossible to recover $\mu^*$ from the action sequence alone.

### B.1    PROOF FOR LINUCB ALGORITHM

We prove by induction on $t$ that the distribution of $a_1, \ldots, a_t$ is independent of $\theta_d^*$.

#### B.1.1    BASE CASE ($t = 1$)

At $t = 1$, the algorithm initializes $V_0 = I_d$ and $b_0 = \mathbf{0}_d$, so $\hat{\theta}_0 = V_0^{-1} b_0 = \mathbf{0}$. The UCB score for an action $a$ is:

$$a^\top \hat{\theta}_0 + \sqrt{\beta_1} \|a\|_{V_0^{-1}} = \sqrt{\beta_1} \|a\|_2.$$

Since $a_d = 1$ for all $a \in \mathcal{A}$, we have $\|a\|_2 = \sqrt{\|a_{1:d-1}\|^2 + 1}$, which depends only on the first $d-1$ dimensions of $a$. Thus, the action selection $a_1 = \arg\max_{a \in \mathcal{A}} \sqrt{\beta_1} \|a\|_2$ is independent of $\theta_d^*$. Therefore, the distribution of $a_1$ is independent of $\theta_d^*$.

### B.1.2 INDUCTIVE STEP

Assume that for steps 1 to $t-1$, the distribution of $a_1, \ldots, a_{t-1}$ is independent of $\theta_d^*$. Consider step $t$. The history $H_t = (a_1, r_1, \ldots, a_{t-1}, r_{t-1})$ is available. The algorithm computes:

$$V_{t-1} = I_d + \sum_{s=1}^{t-1} a_s a_s^\top,$$

$$b_{t-1} = \sum_{s=1}^{t-1} r_s a_s,$$

$$\hat{\theta}_{t-1} = V_{t-1}^{-1} b_{t-1}.$$

The action $a_t$ is chosen as:

$$a_t = \arg\max_{a \in \mathcal{A}} \left[ a^\top \hat{\theta}_{t-1} + \sqrt{\beta_t} \|a\|_{V_{t-1}^{-1}} \right].$$

By the inductive hypothesis, the distribution of $a_1, \ldots, a_{t-1}$ is independent of $\theta_d^*$, so the distribution of $V_{t-1}$ is also independent of $\theta_d^*$. The reward $r_s = \langle \theta^*, a_s \rangle + \eta_s = \langle \theta_{1:d-1}^*, a_s^{(1)} \rangle + \theta_d^* + \eta_s$, where $a_s^{(1)}$ denotes the first $d-1$ dimensions of $a_s$. If $\theta_d^*$ is changed to $\theta_d^* + c$, all rewards $r_s$ are shifted by $c$. In linear regression, shifting the response variable by a constant affects only the intercept estimate (here, $\hat{\theta}_{t-1,d}$) but not the slope estimates ($\hat{\theta}_{t-1,1:d-1}$). Thus, the distribution of $\hat{\theta}_{t-1,1:d-1}$ is independent of $\theta_d^*$.

Now, the UCB score for action $a$ is:

$$a^\top \hat{\theta}_{t-1} + \sqrt{\beta_t} \|a\|_{V_{t-1}^{-1}} = \langle \hat{\theta}_{t-1,1:d-1}, a^{(1)} \rangle + \hat{\theta}_{t-1,d} + \sqrt{\beta_t} \|a\|_{V_{t-1}^{-1}}.$$

Since $\hat{\theta}_{t-1,d}$ is constant for all $a$, it does not affect the $\arg\max$. The term $\|a\|_{V_{t-1}^{-1}}$ depends on $V_{t-1}$, whose distribution is independent of $\theta_d^*$. Therefore, the action selection $a_t$ depends only on $\langle \hat{\theta}_{t-1,1:d-1}, a^{(1)} \rangle$ and $\|a\|_{V_{t-1}^{-1}}$, both of which are independent of $\theta_d^*$. Hence, the conditional distribution of $a_t$ given $H_t$ is independent of $\theta_d^*$.

By induction, the entire action sequence $a_1, \ldots, a_T$ has a distribution independent of $\theta_d^*$.

### B.2 PROOF FOR PHASED ELIMINATION ALGORITHM

The Phased Elimination algorithm operates in phases. We show that the action sequence distribution is independent of $\theta_d^*$ by considering the phase-by-phase updates.

Let $\ell$ denote the phase index. The key step is the elimination condition:

$$\mathcal{A}_{\ell+1} = \left\{ a \in \mathcal{A}_\ell : \max_{b \in \mathcal{A}_\ell} \langle \hat{\theta}_\ell, b - a \rangle \leq 2\varepsilon_\ell \right\}.$$

For any actions $a, b \in \mathcal{A}_\ell$, we have:

$$\langle \hat{\theta}_\ell, b - a \rangle = \langle \hat{\theta}_{\ell,1:d-1}, b^{(1)} - a^{(1)} \rangle,$$

because $b_d - a_d = 0$. Thus, the elimination decision depends only on $\hat{\theta}_{\ell,1:d-1}$.

The estimate $\hat{\theta}_\ell$ is computed via linear regression on the rewards collected in phase $\ell$. The rewards $r_s = \langle \theta^*, a_s \rangle + \eta_s = \langle \theta_{1:d-1}^*, a_s^{(1)} \rangle + \theta_d^* + \eta_s$. A change in $\theta_d^*$ shifts all rewards by a constant, which affects only the intercept estimate $\hat{\theta}_{\ell,d}$ but not the slope estimates $\hat{\theta}_{\ell,1:d-1}$. Therefore, the distribution of $\hat{\theta}_{\ell,1:d-1}$ is independent of $\theta_d^*$.

The G-optimal design $\pi_\ell$ and the sample counts $n_\ell(a)$ depend only on the action set $\mathcal{A}_\ell$, which is updated based on the elimination condition. Since the elimination condition is independent of $\theta_d^*$, the action set $\mathcal{A}_\ell$ and thus the sampling distribution are also independent of $\theta_d^*$.

By induction over phases, the entire action sequence distribution is independent of $\theta_d^*$.

## C  PROOF OF THEOREM 1

### C.1  STEP 1: INFORMATION-THEORETIC LOWER BOUND

We now prove the lower bound for any estimator function $\Psi$. **Note:** We establish the lower bound for an estimator that observes the full trajectory $\tau = (a_1, r_1, \ldots, a_T, r_T)$. The estimator in the theorem definition observes only the action sequence $(a_1, \ldots, a_T)$, which is a subset of the information in $\tau$.

**Construction of Instances:** Fix an action $a \in \mathcal{A}$. Let $\mathcal{M}$ be the instance with parameter $\theta$. We construct an alternative instance $\mathcal{M}'_a$ with parameter $\theta'_a$ defined as:

$$\theta'_a = \theta + \frac{\overline{V}^{-1} a}{\|a\|_{\overline{V}^{-1}}}, \tag{1}$$

where $\overline{V} = I + \mathbb{E}_\theta[\sum_{t=1}^T a_t a_t^\top]$. The separation between the parameters along the direction $a$ is $|a^\top(\theta - \theta'_a)| = \|a\|_{\overline{V}^{-1}}$.

**Change of Measure:** Let $\mathbb{P}_\theta$ and $\mathbb{P}_{\theta'_a}$ denote the probability distributions over trajectories $\tau$ induced by $\theta$ and $\theta'_a$, respectively.

The trajectory density is:

$$\mathbb{P}_\theta(\tau) = \pi(a_1) \prod_{t=2}^T \pi(a_t \mid h_{t-1}) \prod_{t=1}^T \phi(r_t - a_t^\top \theta)$$

where:

- $\pi(\cdot)$ is the action selection policy
- $\phi(x) = (2\pi)^{-1/2} e^{-x^2/2}$ is standard Gaussian density
- $h_{t-1} = (a_1, r_1, \ldots, a_{t-1}, r_{t-1})$ is history

The log-likelihood ratio is:

$$\log \frac{d\mathbb{P}_\theta(\tau)}{d\mathbb{P}_{\theta'_a}(\tau)} = \log \frac{\pi(a_1)}{\pi(a_1)} + \sum_{t=2}^T \log \frac{\pi(a_t \mid h_{t-1})}{\pi(a_t \mid h_{t-1})} + \sum_{t=1}^T \log \frac{\phi(r_t - a_t^\top \theta)}{\phi(r_t - a_t^\top \theta'_a)},$$

where $\phi$ is the standard Gaussian density. Crucially, since the demonstrator's policy $\pi$ depends only on the history $h_{t-1}$, the policy terms cancel out.

The KL divergence is determined solely by the reward distributions:

$$D_{\mathrm{KL}}(\mathbb{P}_\theta \| \mathbb{P}_{\theta'_a}) = \mathbb{E}_{\mathbb{P}_\theta}\left[ \sum_{t=1}^T \frac{1}{2} \left( (r_t - a_t^\top \theta'_a)^2 - (r_t - a_t^\top \theta)^2 \right) \right]$$

$$= \frac{1}{2} \mathbb{E}_{\mathbb{P}_\theta}\left[ \sum_{t=1}^T (a_t^\top(\theta - \theta'_a))^2 \right].$$

Substituting $\theta - \theta'_a = -\overline{V}^{-1} a / \|a\|_{\overline{V}^{-1}}$, and using the identity $\mathbb{E}_{\mathbb{P}_\theta}[\sum a_t a_t^\top] = \overline{V} - I$, we obtain:

$$D_{\mathrm{KL}}(\mathbb{P}_\theta \| \mathbb{P}_{\theta'_a}) = \frac{1}{2\|a\|_{\overline{V}^{-1}}^2} (\overline{V}^{-1} a)^\top (\overline{V} - I)(\overline{V}^{-1} a)$$

$$= \frac{1}{2} \left( 1 - \frac{a^\top \overline{V}^{-2} a}{a^\top \overline{V}^{-1} a} \right) \leq \frac{1}{2} \quad .$$

Using Pinsker's inequality, the total variation distance is bounded by $\|\mathbb{P}_\theta - \mathbb{P}_{\theta'_a}\|_{\mathrm{TV}} \leq \sqrt{\frac{1}{2} D_{\mathrm{KL}}} \leq \frac{1}{2}$.

**Minimax Probability of Error:** Let $\hat{\theta} = \Psi(a_1, ..., a_T)$ be the estimate. We consider the event that the estimation error is large:

$$\mathcal{E}_\nu = \left\{ |a^\top(\hat{\theta} - \nu)| \geq \frac{1}{2}\|a\|_{\overline{V}^{-1}} \right\}, \nu \in \{\theta, \theta'_a\}.$$

By the triangle inequality, if $|a^\top(\hat{\theta} - \theta)| < \frac{1}{2}\|a\|_{\overline{V}^{-1}}$ and $|a^\top(\hat{\theta} - \theta'_a)| < \frac{1}{2}\|a\|_{\overline{V}^{-1}}$, then $|a^\top(\theta - \theta'_a)| < \|a\|_{\overline{V}^{-1}}$, which contradicts the construction of $\theta'_a$. Thus:

$$\mathbb{P}_\theta(\mathcal{E}_\theta) + \mathbb{P}_{\theta'_a}(\mathcal{E}_{\theta'_a}) \geq 1 - \|\mathbb{P}_\theta - \mathbb{P}_{\theta'_a}\|_{\text{TV}} \geq 1 - \frac{1}{2} = \frac{1}{2}.$$

Therefore, $\max_{\nu \in \{\theta, \theta'_a\}} \mathbb{P}_\nu(\mathcal{E}_\nu) \geq \frac{1}{4}$.

## C.2 STEP 2: AVAILABILITY OF THE APPROXIMATION $\mu$

We now prove the moreover part of the theorem: when the demonstrator algorithm employs Phased Elimination or LinUCB with $\delta_{\text{PE}} \leq 1/T$ or $\delta_{\text{UCB}} \leq 1/T$, the lower bound holds even if an approximation of the best reward $\mu$ satisfying $\mu \in [\mu^*, \mu^* + \kappa]$ with $\kappa = O\left(d \log T/\sqrt{T}\right)$ is given to the estimator.

First, note that with high probability, the cumulative regret $R_T$ of the demonstrator algorithm is $O(d\sqrt{T} \log T)$. Specifically, for Phased Elimination or LinUCB with $\delta \leq 1/T$, we have with probability at least $1 - 1/T$, $R_T \leq Cd\sqrt{T} \log T$ for some constant $C$.

The estimator can compute the average reward $\hat{\mu}_T = \frac{1}{T}\sum_{t=1}^T r_t$. Since the noise is Gaussian, by Gaussian concentration, for any $\delta > 0$,

$$\mathbb{P}\left( |\hat{\mu}_T - \mathbb{E}[\hat{\mu}_T]| \geq \sqrt{\frac{2\log(2/\delta)}{T}} \right) \leq \delta.$$

Setting $\delta = 1/T$, we have with probability at least $1 - 1/T$,

$$|\hat{\mu}_T - \mathbb{E}[\hat{\mu}_T]| \leq \sqrt{\frac{2\log(2T)}{T}} = O\left( \sqrt{\frac{\log T}{T}} \right).$$

Moreover, since $\mathbb{E}[\hat{\mu}_T] = \frac{1}{T}\sum_{t=1}^T a_t^\top\theta$ and $\mu^* = \max_{a \in \mathcal{A}} a^\top\theta$, we have

$$|\mathbb{E}[\hat{\mu}_T] - \mu^*| \leq \frac{R_T}{T}.$$

Thus, with probability at least $1 - 1/T$ (from the regret bound),

$$|\mathbb{E}[\hat{\mu}_T] - \mu^*| \leq O\left( \frac{d \log T}{\sqrt{T}} \right).$$

By the union bound, with probability at least $1 - 2/T$, both events hold, and hence

$$|\hat{\mu}_T - \mu^*| \leq |\hat{\mu}_T - \mathbb{E}[\hat{\mu}_T]| + |\mathbb{E}[\hat{\mu}_T] - \mu^*| \leq O\left( \sqrt{\frac{\log T}{T}} \right) + O\left( \frac{d \log T}{\sqrt{T}} \right) = O\left( \frac{d \log T}{\sqrt{T}} \right).$$

The estimator then sets $\mu = \hat{\mu}_T + c$ for some constant $c = O(d \log T/\sqrt{T})$ to ensure that with high probability, $\mu \geq \mu^*$ and $\mu \leq \mu^* + \kappa$ with $\kappa = O(d \log T/\sqrt{T})$. Define the event $E$ that $\mu \in [\mu^*, \mu^* + \kappa]$. Then $\mathbb{P}(E) \geq 1 - 2/T$.

At last, we account for the failure probability of event $E$ (the approximation of $\mu$). The source establishes that conditioned on $E$, the max probability is at least $1/4$, the unconditional probability is:

$$\max_{\nu \in \{\theta, \theta'_a\}} \mathbb{P}_\nu(\mathcal{E}_\nu) \geq \frac{1}{4}\left( 1 - \frac{2}{T} \right).$$

For $T \geq 3$, we have $(1 - 2/T) \geq 1/3$. Thus:

$$\max_{\nu \in \{\theta, \theta'_a\}} \mathbb{P}_\nu\left( |a^\top(\hat{\theta} - \nu)| \geq \frac{1}{2}\|a\|_{\overline{V}^{-1}} \right) \geq \frac{1}{12}.$$

$\square$

## D   MISSING PROOFS IN SECTION 5

**Lemma 4.** *For all phase $\ell$ and all actions $a \in \mathcal{A}_\ell$, for $\nu \in (0,1)$, we can set $\delta_{\mathsf{PE}}$ so that with probability at least $1 - \dfrac{1}{T}$, the following holds:*

$$\langle \hat{\theta}_\ell - \theta^*, a \rangle \leq \nu \epsilon_\ell$$

*Proof.* In phased elimination algorithm, with probability at least $1 - \delta_{\mathsf{PE}}$, we have $\langle \hat{\theta}_\ell - \theta^*, a \rangle \leq \epsilon_\ell$.
Set $\delta_{\mathsf{PE}}$ such that for all $\ell \in [1, L]$,

$$\frac{8d\pi_\ell(a)}{\epsilon_\ell^2} \left( d \log 6 + \log \left( \frac{\ell(\ell+1)}{\delta_{\mathsf{PE}}} \right) \right) \geq \frac{8d\pi_\ell(a)}{(\nu \epsilon_\ell)^2} \left( d \log 6 + \log \left( \frac{\ell(\ell+1)}{1/T} \right) \right)$$

Then, with probability at least $1 - \dfrac{1}{T}$, we have:

$$\langle \hat{\theta}_\ell - \theta^*, a \rangle \leq \nu \epsilon_\ell$$

$\square$

### D.1   PROOF OF THEOREM 2

*Proof.* **Setup.** Let $L$ be the number of phases, $\{\pi_\ell\}_{\ell=1}^L$ the per-phase designs, and $\mathrm{Supp}(\pi_\ell)$ denote its support. The empirical regularized design matrix is:

$$\hat{V} := I + \sum_{\ell=1}^L \sum_{a \in \mathrm{Supp}(\pi_\ell)} n_\ell(a) \, aa^\top,$$

where $n_\ell(a) = \frac{8d\,\pi_\ell(a)}{\epsilon_\ell^2} \, p_\ell$ and $p_\ell := d \log 6 + \log \left( \frac{\ell(\ell+1)}{\delta_{\mathsf{PE}}} \right)$.

**Feasibility.** Fix any round $\ell \geq 2$ and $a \in \mathcal{A}_\ell$. By Lemma 2 and the Phased Elimination selection rule:

$$a^\top \theta^* \geq a^\top \hat{\theta}_{\ell-1} - \epsilon_{\ell-1} \geq \left( \max_{b \in \mathcal{A}_{\ell-1}} b^\top \hat{\theta}_{\ell-1} \right) - 2\epsilon_{\ell-1}.$$

Since $b^\top \hat{\theta}_{\ell-1} \geq b^\top \theta^* - \epsilon_{\ell-1}$, we get $a^\top \theta^* \geq \mu^* - 4\epsilon_{\ell-1} - \epsilon_{\ell-1} = \mu^* - 8\epsilon_\ell$. Using $\mu^* \geq \mu - \kappa$, we obtain $a^\top \theta^* \geq \mu - 8\epsilon_\ell - \kappa$. Also $a^\top \theta^* \leq \mu^* \leq \mu$. Thus, $\theta^*$ satisfies the constraint system with $\kappa$-slack, making it feasible.

**Decomposition.** Let $\hat{\theta}$ and $\hat{\theta}'$ be feasible solutions with slack $\kappa$ and $0$, respectively. Define:

$$\Delta := \hat{\theta} - \theta^*, \qquad \Delta' := \hat{\theta}' - \theta^*, \qquad \delta := \hat{\theta} - \hat{\theta}'.$$

The error decomposes exactly as: $|\langle \Delta, a \rangle| \leq |\langle \Delta', a \rangle| + |\langle \delta, a \rangle|$.

**Statistical Error** ($|\langle \Delta', a \rangle|$)**.** For $a \in \mathrm{Supp}(\pi_\ell)$, constraints with $\kappa = 0$ yield $|\langle \Delta', a \rangle| \leq 8\epsilon_\ell$. Using $\|\Delta'\|_2^2 \leq 4d$:

$$\|\Delta'\|_{\hat{V}}^2 \leq \|\Delta'\|_2^2 + \sum_{\ell=1}^L \sum_{a \in \mathrm{Supp}(\pi_\ell)} n_\ell(a) \, (8\epsilon_\ell)^2$$

$$= 4d + \sum_{\ell=1}^L \sum_{a \in \mathrm{Supp}(\pi_\ell)} \left( \frac{8d\pi_\ell(a)}{\epsilon_\ell^2} p_\ell \right) \cdot 64\epsilon_\ell^2$$

$$= 4d + 512\,d \sum_{\ell=1}^L p_\ell$$

$$= O\left( d^2 L \log \frac{L}{\delta_{\mathsf{PE}}} \right) = O\left( d^2 \log T \log \frac{L}{\delta_{\mathsf{PE}}} \right).$$

By Cauchy–Schwarz:

$$|\langle \Delta', a\rangle| \le \|\Delta'\|_{\hat{V}} \|a\|_{\hat{V}^{-1}} = O\left(\sqrt{d^2 \log T \log \frac{L}{\delta_{\mathsf{PE}}}}\right) \|a\|_{\hat{V}^{-1}}. \tag{2}$$

**Perturbation Error** ($|\langle \delta, a\rangle|$)**.** For any $b \in \mathrm{Supp}(\pi_1)$, we have $|\langle \delta, b\rangle| \le \kappa$. Using the G-optimal phase-1 design matrix $V(\pi_1)$:

$$\|\delta\|^2_{V(\pi_1)} = \sum_{b \in \mathrm{Supp}(\pi_1)} \pi_1(b)\, \langle \delta, b\rangle^2 \le \sum_b \pi_1(b)\, \kappa^2 = \kappa^2,$$

which implies $\|\delta\|_{V(\pi_1)} \le \kappa$. By the G-optimality property, $\|a\|_{V(\pi_1)^{-1}} \le \sqrt{d}$ for all $a$. Applying Cauchy–Schwarz in the $V(\pi_1)$ inner product:

$$|\langle \delta, a\rangle| \le \|\delta\|_{V(\pi_1)} \|a\|_{V(\pi_1)^{-1}} \le \sqrt{d}\,\kappa. \tag{3}$$

**Conclusion.** Combining equation 2 and equation 3:

$$|\langle \Delta, a\rangle| \le O\left(\sqrt{d^2 \log T \log \frac{L}{\delta_{\mathsf{PE}}}}\right) \|a\|_{\hat{V}^{-1}} + \sqrt{d}\,\kappa.$$

$\square$

### D.2 Proof of Theorem 3

We define $V^* = I + \sum_{\ell=1}^L V_\ell^*$ and $\mathcal{A}_\ell^*(\ell \in [1, L])$ by algorithm 7:

---

**Algorithm 7** Algorithm for Phased Elimination proof

---

1: $\ell \leftarrow 0, \mathcal{A}_1^* \leftarrow \mathcal{A}$
2: **while** Number of rounds $\le T$ **do**
3: $\quad \varepsilon_\ell \leftarrow 2^{-\ell}$
4: $\quad \pi_\ell^* \leftarrow$ G-OptimalDesign$(\mathcal{A}_\ell^*)$
5: $\quad$ **for** each $a \in \mathcal{A}_\ell^*$ **do**
6: $\quad\quad n_\ell^*(a) \leftarrow \frac{8d\pi_\ell^*(a)}{\epsilon_\ell^2}\left(d \log 6 + \log\left(\frac{\ell(\ell+1)}{\delta_{\mathsf{PE}}}\right)\right)$
7: $\quad$ **end for**
8: $\quad$ Play each $a \in \mathcal{A}_\ell$ exactly $n_\ell(a)$ times
9: $\quad V_\ell^* \leftarrow \sum_{a \in \mathcal{A}_\ell^*} n_\ell^*(a)aa^\top$
10: $\quad \mathcal{A}_{\ell+1}^* \leftarrow \{a \in \mathcal{A}_\ell^* : \max_{b \in \mathcal{A}_\ell}\langle \theta^*, b - a\rangle \le 2\varepsilon_\ell\}$
11: $\quad \ell \leftarrow \ell + 1$
12: **end while**

---

**Lemma 5.** *For the matrices*

$$V_\ell^* = \sum_{a \in \mathcal{A}_\ell^*} n_\ell^*(a)\, aa^\top, \quad n_\ell^*(a) = \frac{8d\,\pi_\ell^*(a)}{\varepsilon_\ell^2}\left(d \log 6 + \log \frac{\ell(\ell+1)}{\delta_{\mathrm{PE}}}\right),$$

*with $\varepsilon_\ell = 2^{-\ell}$ and $\mathcal{A}_{\ell+1}^* \subseteq \mathcal{A}_\ell^*$, for any integer $\alpha \ge 1$ we have*

$$V_{\ell-\alpha}^* \preceq C_\alpha V_{\ell+\alpha}^*, \qquad C_\alpha = O(16^\alpha).$$

*Proof.* By definition,

$$V_\ell^* = \frac{8d}{\varepsilon_\ell^2}\left(d \log 6 + \log \frac{\ell(\ell+1)}{\delta_{\mathrm{PE}}}\right) \sum_{a \in \mathcal{A}_\ell^*} \pi_\ell^*(a)\, aa^\top.$$

Since $\varepsilon_{\ell-\alpha}^{-2}/\varepsilon_{\ell+\alpha}^{-2} = (\varepsilon_{\ell+\alpha}/\varepsilon_{\ell-\alpha})^2 = 16^\alpha$ and the logarithmic term changes by at most a constant factor $c_{\log}$, the scalar coefficients satisfy

$$\frac{\frac{8d}{\varepsilon_{\ell-\alpha}^2}\left(d \log 6 + \log \frac{(\ell-\alpha)(\ell-\alpha+1)}{\delta_{\mathrm{PE}}}\right)}{\frac{8d}{\varepsilon_{\ell+\alpha}^2}\left(d \log 6 + \log \frac{(\ell+\alpha)(\ell+\alpha+1)}{\delta_{\mathrm{PE}}}\right)} \le 16^\alpha c_{\log}.$$

Moreover, since $\mathcal{A}_{\ell+\alpha}^* \subseteq \mathcal{A}_{\ell-\alpha}^*$, the moment matrices satisfy

$$\sum_{a \in \mathcal{A}_{\ell-\alpha}^*} \pi_{\ell-\alpha}^*(a)\, aa^\top \preceq c_\pi \sum_{a \in \mathcal{A}_{\ell+\alpha}^*} \pi_{\ell+\alpha}^*(a)\, aa^\top,$$

for some constant $c_\pi \geq 1$. Combining the two relations yields

$$V_{\ell-\alpha}^* \preceq 16^\alpha c_{\log} c_\pi\, V_{\ell+\alpha}^*,$$

and the constants are absorbed into $C_\alpha = O(16^\alpha)$. $\qquad\square$

For any $a \in \mathcal{A}$, define $r = \langle \theta^*, \mu^* - a \rangle, \gamma = \log_2(\frac{2}{r})$.

In algorithm 7, $a$ is eliminated within $(\gamma - 1, \gamma + 1)$ rounds.

Set $\nu = \dfrac{1}{1+c}$, and set $h = 2 - 2\nu$. Then:

$$r = h \cdot 2^{-(\gamma + \log_2 h/2)}$$

For $\ell \leq \gamma + \log_2 h/2, \forall a' \in \mathcal{A}_\ell$, by Lemma 4:

$$\langle \hat{\theta}_l, a' - a \rangle \leq |\langle \hat{\theta}_\ell - \theta_*, a' \rangle| + r + |\langle \hat{\theta}_\ell - \theta_*, a \rangle| \leq 2\nu\epsilon_\ell + h\epsilon_\ell = 2\epsilon_\ell$$

Thus $a$ is eliminated after round $\lfloor \gamma + \log_2 h/2 \rfloor$. Set $\alpha = \lceil -\log_2 h/2 + 1 \rceil$.

Furthermore, it's obvious that $a$ is eliminated before round $\lceil \gamma + 2 \rceil$, then $a$ is eliminated within $(\gamma - \alpha + 1, \gamma + \alpha - 1)$ rounds.

For $\ell \in [\alpha + 1, T], a \in A_\ell$, we have $a \in A_{\ell-\alpha}^*$ (otherwise $\gamma < l - \alpha$ implies $l > \gamma + \alpha$). Hence $A_l \subseteq A_{l-\alpha}^*$.

Define $V_\ell(\pi_\ell) = \sum_{a \in \mathcal{A}_\ell} \pi_\ell(a) aa^\top$, and $V_\ell^*$ goes similarly

By Lemma 1:

$$V_\ell(\pi_\ell) \preceq d \cdot V_{\ell-\alpha}^*(\pi_{\ell-\alpha}^*)$$

Substitute expressions:

$$V_\ell = \frac{2d}{\epsilon_\ell^2} \log \frac{6^d \ell(\ell+1)}{\delta_{\mathsf{PE}}} V_l(\pi_l)$$

$$V_{l-\alpha}^* = \frac{2d}{\epsilon_{\ell-\alpha}^2} \log \frac{6^d(\ell-\alpha)(\ell-\alpha+1)}{\delta_{\mathsf{PE}}} V_{l-\alpha}^*(\pi_{l-\alpha}^*)$$

Obtain:

$$V_l \preceq d \cdot \frac{\log \frac{6^d \ell(\ell+1)}{\delta_{\mathsf{PE}}}}{\log \frac{6^d(\ell-\alpha)(\ell-\alpha+1)}{\delta_{\mathsf{PE}}}} \cdot 4^\alpha \cdot V_{\ell-\alpha}^*$$

$$\preceq d \cdot \underbrace{\frac{\log(l(l+1))}{\log((l-\alpha)(l-\alpha+1))}}_{\tau_1(l)} \cdot 4^\alpha \cdot V_{l-\alpha}^*$$

let $\tau_1 = \sup_{\ell \geq \alpha+1} \tau_1(\ell) < \infty$ obtains:

$$V_l \preceq d \cdot 4^\alpha \cdot \tau_1 \cdot V_{l-\alpha}^*, \forall l \in [\alpha+1, T]$$

Similarly:

$$V_l \succeq d^{-1} \cdot 4^{-\alpha} \cdot \tau_2 \cdot V_{l+\alpha}^*, \forall l \in [1, T-\alpha]$$

for a constant $\tau_2$.

For $\ell \in [1, \alpha]$:

$$V_\ell \preceq d \cdot 4^{\ell-1} \cdot \frac{\log(\ell(\ell+1))}{\log 2} \cdot V_1^*$$

Combining terms:

$$\hat{V} = I + \sum_{l=1}^{\alpha} V_l + \sum_{l=\alpha+1}^{L} V_l$$

$$\preceq I + \alpha \cdot d \cdot 4^{\alpha-1} \cdot \frac{\log(\alpha(\alpha+1))}{\log 2} \cdot V_1^*$$

$$+ d \cdot 4^{\alpha} \cdot \tau_1 \sum_{l=1}^{L-\alpha} V_l^*$$

$$\preceq C_1 d V^*$$

The lower bound $\hat{V} \succeq C_2 d^{-1} V^*$ follows similarly, where $C1, C2$ are both constants.

Substituting Lemma 4 we have:

$$\overline{V} \preceq C_1 d V^* + T^{-1} \cdot (T+1) I$$

Recall that $\hat{V} \succeq C_2 d^{-1} V^*$, which completes the proof.

### D.3 PROOF OF THEOREM 4

*Proof.* **Setup.** We use the notation from the main text. Let $\hat{V}$ be the empirical regularized design matrix:

$$\hat{V} := I + \sum_{\ell=1}^{L} \sum_{a \in \mathrm{Supp}(\pi_\ell)} n_\ell(a) \, a a^\top,$$

where $n_\ell(a) = \frac{8d \, \pi_\ell(a)}{\epsilon_\ell^2} p_\ell$. We choose $C$ such that $C \cdot \sqrt{\frac{d^2}{t}} \geq 2^{-\ell_t}$.

**Feasibility.** By the retention rule in Phased Elimination, for $a \in \mathcal{A}_\ell$, $a^\top \theta^* \geq \mu^* - 8\epsilon_\ell$. With the $\kappa$-slack definition $\mu^* \geq \mu - \kappa$, we have $a^\top \theta^* \geq \mu - 8\epsilon_\ell - \kappa$. Also $a^\top \theta^* \leq \mu^* \leq \mu$. Since $2^{-\ell_t} = O(\sqrt{d^2/t})$ by construction, $\theta^*$ is a feasible solution to the constraint system with $\kappa$-slack.

**Decomposition.** Let $\hat{\theta}$ and $\hat{\theta}'$ be feasible solutions with slack $\kappa$ and $0$, respectively. Define the error vectors (assuming $\|\hat{\theta}\|_2, \|\hat{\theta}'\|_2 \leq \sqrt{d}$):

$$\Delta := \hat{\theta} - \theta^*, \qquad \Delta' := \hat{\theta}' - \theta^*, \qquad \delta := \hat{\theta} - \hat{\theta}'.$$

The total error decomposes as $|\langle \Delta, a \rangle| \leq |\langle \Delta', a \rangle| + |\langle \delta, a \rangle|$.

**Statistical Error** ($|\langle \Delta', a \rangle|$)**.** Since $\hat{\theta}'$ is feasible with $\kappa = 0$, the sampled action $a_t$ at time $t$ satisfies $|\langle \Delta', a_t \rangle| \leq C\sqrt{d^2/t}$. Expanding the quadratic form and using the bounds (similar to the Phased Elimination case in Theorem 2 and the analysis of $\sum d^2/t$):

$$\|\Delta'\|_{\hat{V}}^2 = \|\Delta'\|_2^2 + \sum_{t=1}^{T} \langle \Delta', a_t \rangle^2 \leq 4d + \sum_{t=1}^{T} C^2 \frac{d^2}{t} = O(C^2 d^2 \log T).$$

By Cauchy–Schwarz:

$$|\langle \Delta', a \rangle| \leq \|\Delta'\|_{\hat{V}} \|a\|_{\hat{V}^{-1}} = O\left(\sqrt{C^2 d^2 \log T}\right) \|a\|_{\hat{V}^{-1}}. \tag{4}$$

By Lemma 2, $C = O(\sqrt{\log(L/\delta_{\mathsf{PE}})})$, leading to the final form: $|\langle \Delta', a \rangle| = O\left(\sqrt{d^2 \log T \log\left(\frac{L}{\delta_{\mathsf{PE}}}\right)}\right) \|a\|_{\hat{V}^{-1}}$.

**Perturbation Error** ($|\langle \delta, a \rangle|$)**.** The analysis is identical to Theorem 2. Constraints imply $|\langle \delta, a_t \rangle| \leq \kappa$. Using the G-optimal phase-1 design $V(\pi_1)$:

$$\|\delta\|_{V(\pi_1)}^2 \leq \kappa^2.$$

Since $\|a\|_{V(\pi_1)^{-1}} \le \sqrt{d}$, by Cauchy–Schwarz:

$$|\langle \delta, a \rangle| \le \sqrt{d}\,\kappa. \tag{5}$$

**Conclusion.** Combining equation 4 and equation 5:

$$|\langle \Delta, a \rangle| \le O\left(\sqrt{d^2 \log T \log\left(\tfrac{L}{\delta_{\mathsf{PE}}}\right)}\right) \|a\|_{\hat{V}^{-1}} + \sqrt{d}\,\kappa.$$

$\square$

## E    MISSING PROOFS IN SECTION 6

**Lemma 6.** *For any round $t$ and all actions $a$, for $\nu \in (0,1)$, we can set $\delta_{\mathsf{UCB}}$ so that with probability at least $1 - \dfrac{1}{T}$, the following holds:*

$$\langle \theta^*, a \rangle + \nu \sqrt{\beta_t} \|a\|_{V_{t-1}^{-1}} \le \mathrm{UCB}_t(a) \le \langle \theta^*, a \rangle + 2\sqrt{\beta_t} \|a\|_{V_{t-1}^{-1}}$$

The lemmas below all happens under the event of Lemma 6

*Proof.* With probability at least $1 - \delta_{\mathsf{UCB}}$, we have $\theta^* \in \{\theta \in \mathbb{R}^d \mid \|\theta - \hat{\theta}_{t-1}\|_{V_{t-1}} \le \sqrt{\beta_t}\}$.

Set $\delta_{\mathsf{UCB}}$ such that:

$$(1 - \nu)\sqrt{\beta_t} \ge \sqrt{d} + \sqrt{2\log(T) + d \log \frac{d+t}{d}}$$

Then, with probability at least $1 - \dfrac{1}{T}$, we have:

$$\theta^* \in \{\theta \in \mathbb{R}^d \mid \|\theta - \hat{\theta}_{t-1}\|_{V_{t-1}} \le (1 - \nu)\sqrt{\beta_t}\}$$

The bound on $\mathrm{UCB}_t(a)$ follows from the definition of the upper confidence bound and the concentration inequality.

$\square$

**Lemma 7.** *In the UCB algorithm, define:*

$$S = \left\{ t \in [1, T] \mid \Delta_{a_t} \le \frac{2C_R d \log T}{\sqrt{T}} \right\}, \quad Q = \left\{ a \in A \mid \Delta_a \le \frac{2C_R d \log T}{\sqrt{T}} \right\}$$

*There exist constants $c_1 = 0.5$ and $c_2 = \dfrac{2C_R + 2}{\nu}$ such that:*

1. *$|S| \ge c_1 T$*

2. *For all $a \in Q$, $\|a\|_{\hat{V}^{-1}} \le \dfrac{c_2 d \log T}{\sqrt{T}}$*

*Proof.* 1. If $|S| < c_1 T$, then the regret would be:

$$R_T > (T - c_1 T) \cdot \frac{2C_R d \log T}{\sqrt{T}} \ge C_R d\sqrt{T} \log T$$

which contradicts the regret bound in Lemma 3. Hence, $|S| \ge c_1 T$.

2. For any $t \in S$, if for all $a \in Q$, $\|a\|_{V_{t-1}^{-1}} \le \dfrac{c_2 d \log T}{\sqrt{T}}$, then since $\hat{V}^{-1} \preceq V_{t-1}^{-1}$, the result follows immediately.

Now, let $c_3 = 2$. Suppose for contradiction that there exists $a \in Q$ such that $\|a\|_{\hat{V}^{-1}} > \dfrac{c_2 d \log T}{\sqrt{T}}$. Consider two cases:

**Case 1:** There exists $b \in Q$ such that $\|b\|_{V_{t-1}^{-1}} \leq \dfrac{c_3 d \log T}{\sqrt{T}}$. Then by Lemma 6:

$$\text{UCB}_t(a) - \text{UCB}_t(b) \geq \sqrt{\beta_t} \left( \nu \|a\|_{V_{t-1}^{-1}} - 2\|b\|_{V_{t-1}^{-1}} \right) - \frac{2C_R d \log T}{\sqrt{T}}$$

Substituting the bounds:

$$> \frac{d \log T}{\sqrt{T}} \left( (\nu c_2 - c_3) \sqrt{\beta_t} - 2C_R \right) > 0$$

Thus, $b$ would not be selected.

**Case 2:** For all $b \in Q$, $\|b\|_{V_{t-1}^{-1}} > \dfrac{c_3 d \log T}{\sqrt{T}}$.

As for both Case 1 and Case 2, the selected action $a_t$ satisfies $\|a_t\|_{V_{t-1}^{-1}} > \dfrac{c_3 d \log T}{\sqrt{T}}$.

But then:

$$\sum_{t=1}^{T} \|a_t\|_{V_{t-1}^{-1}}^2 > c_1 T \cdot \left( \frac{c_3 d \log T}{\sqrt{T}} \right)^2 = c_1 c_3^2 d^2 \log^2 T$$

Since $c_1 = 0.5$ and $c_3 = 2$, we have $c_1 c_3^2 = 2$, so:

$$\sum_{t=1}^{T} \|a_t\|_{V_{t-1}^{-1}}^2 > 2d^2 \log^2 T$$

which contradicts the sum bound in Lemma 3 (since $\sum_{t=1}^{T} \|a_t\|_{V_{t-1}^{-1}}^2 \leq 2d \log T$).

Therefore, the assumption is false, and the result holds. $\qquad\square$

**Lemma 8.** *In the UCB algorithm, there exists a constant $c_4 = \dfrac{c_2/C_R + 1}{\nu}$ such that after $T$ rounds, for any action $a$ with $\Delta_a > \dfrac{2C_R d \log T}{\sqrt{T}}$, we have:*

$$\|a\|_{\hat{V}^{-1}} \leq c_4 \Delta_a$$

*Proof.* For any $a$ with $\Delta_a > \dfrac{2C_R d \log T}{\sqrt{T}}$, assume for contradiction that at time $t$, $\|a\|_{V_{t-1}^{-1}} > c_4 \Delta_a$.

For any $x \in Q$, by Lemma 6:

$$\text{UCB}_t(a) - \text{UCB}_t(x) \geq \langle \theta^*, a - x \rangle + \sqrt{\beta_t} \left( \frac{1}{2} \|a\|_{V_{t-1}^{-1}} - 2\|x\|_{V_{t-1}^{-1}} \right)$$

Using $\langle \theta^*, a - x \rangle > -\Delta_a$ and the bounds from Lemma 7:

$$> \sqrt{\beta_t} \left( \nu c_4 \Delta_a - 2 \cdot \frac{c_2 d \log T}{\sqrt{T}} \right) - \Delta_a$$

Substitute $c_4 = \dfrac{c_2/C_R + 1}{\nu}$:

$$\geq \sqrt{\beta_t} \left( (c_2/C_R + 1)\Delta_a - 2 \cdot \frac{c_2 d \log T}{\sqrt{T}} \right) - \Delta_a$$

Since $\Delta_a > \dfrac{2C_R d \log T}{\sqrt{T}}$, we have:

$$> \sqrt{\beta_t} \left( (c_2/C_R + 1)\Delta_a - c_2 \Delta_a/C_R \right) - \Delta_a = \sqrt{\beta_t} \Delta_a - \Delta_a > 0$$

Thus, no action in $Q$ would be selected at time $t$. But by Lemma 7, during $[c_1 T, T]$, at least one action from $Q$ is selected. Hence, there exists some $t \in [c_1 T, T]$ such that $\|a\|_{V_{t-1}^{-1}} \leq c_4 \Delta_a$. Since $\hat{V}^{-1} \preceq V_{t-1}^{-1}$, the result follows. $\qquad\square$

**Lemma 9.** *For any selected action $a_t$ at time $t$ with gap $\Delta_a$, we have:*

$$\|a_t\|_{V_{t-1}^{-1}} \geq \frac{\Delta_a}{2\sqrt{\beta_t}}$$

*Proof.* If action $a$ is selected at time $t$, it must satisfy:

$$\text{UCB}_t(a) \geq \text{UCB}_t(a^*) \Rightarrow \langle \hat{\theta}, a \rangle + \sqrt{\beta_t}\|a\|_{V_{t-1}^{-1}} \geq \langle \hat{\theta}, a^* \rangle + \sqrt{\beta_t}\|a^*\|_{V_{t-1}^{-1}}$$

Let $E(t) = \hat{\theta}_{t-1} - \theta^*$. Rearranging terms:

$$\langle E(t), a - a^* \rangle - \Delta_a + \sqrt{\beta_t}(\|a\|_{V_{t-1}^{-1}} - \|a^*\|_{V_{t-1}^{-1}}) \geq 0 \quad \text{(A)}$$

By Cauchy-Schwarz inequality:

$$|\langle E(t), a - a^* \rangle| \leq \|E(t)\|_{V_{t-1}} \cdot \|a - a^*\|_{V_{t-1}^{-1}} \leq \sqrt{\beta_t}\|a - a^*\|_{V_{t-1}^{-1}}$$

Substituting into (A):

$$\sqrt{\beta_t}\|a - a^*\|_{V_{t-1}^{-1}} - \Delta_a + \sqrt{\beta_t}(\|a\|_{V_{t-1}^{-1}} - \|a^*\|_{V_{t-1}^{-1}}) \geq 0$$

By triangle inequality:

$$\sqrt{\beta_t}(\|a\|_{V_{t-1}^{-1}} + \|a^*\|_{V_{t-1}^{-1}}) - \Delta_a + \sqrt{\beta_t}(\|a\|_{V_{t-1}^{-1}} - \|a^*\|_{V_{t-1}^{-1}}) \geq 0$$

Simplifying:

$$2\sqrt{\beta_t}\|a\|_{V_{t-1}^{-1}} - \Delta_a \geq 0 \Rightarrow \|a\|_{V_{t-1}^{-1}} \geq \frac{\Delta_a}{2\sqrt{\beta_t}}$$

$\square$

Let $L = \min\left\{L \in \mathbb{N}^* \mid \sum_{\ell=1}^{L} d \cdot 4^\ell \geq T\right\}$.

We define $V^* = I + \sum_{\ell=1}^{L} V_\ell^*$ by algorithm 8

---

**Algorithm 8** algorithm for UCB proof

---

1: $\ell \leftarrow 0, \mathcal{A}_1 \leftarrow \mathcal{A}$
2: **while** $\ell \leq L$ **do**
3:     $\pi_\ell \leftarrow$ Approximate G-Optimal Design$(\mathcal{A}_\ell)$
4:     **for** each $a \in \mathcal{A}_\ell$ **do**
5:         $n_\ell(a) \leftarrow d \cdot 4^\ell \cdot \pi(a)$
6:     **end for**
7:     $V_\ell^* \leftarrow \sum_{a \in \mathcal{A}_\ell} n_\ell(a) aa^\top$
8:     $\mathcal{A}_{\ell+1} \leftarrow \left\{a \in \mathcal{A}_\ell : \max_{b \in \mathcal{A}_\ell} \langle \theta^*, b - a \rangle \leq 2^{-\ell+1}\right\}$
9:     $\ell \leftarrow \ell + 1$
10: **end while**

---

Define the partition:

- $P_1 = \{t \in [1, T] \mid \Delta_{a_t} \in (1, \infty)\}$
- $P_L = \{t \in [1, T] \mid \Delta_{a_t} \in [0, 2^{-L+2})\}$
- For $\ell \in [2, L-1]$, $P_\ell = \{t \in [1, T] \mid \Delta_{a_t} \in [2^{-\ell+1}, 2^{-\ell+2})\}$

**Lemma 10.** *For any $\ell \in [1, L-1]$, there exists a constant $c_5 = 2$ such that:*

$$|P_\ell| \leq c_5 \beta_T d \log T \cdot 4^\ell$$

*Proof.* Assume for contradiction that for some $\ell \in [1, L-1]$, $|P_\ell| > c_5 \beta_T d \log T \cdot 4^\ell$.

By Lemma 9, for all $t \in P_\ell$:

$$\|a_t\|_{V_{t-1}^{-1}} \geq \frac{\Delta_{a_t}}{2\sqrt{\beta_t}} \geq \frac{2^{-\ell+1}}{2\sqrt{\beta_T}} = \frac{2^{-\ell}}{\sqrt{\beta_T}}$$

Then:

$$\sum_{t=1}^{T} \|a_t\|_{V_{t-1}^{-1}}^2 > |P_\ell| \cdot \left(\frac{2^{-\ell}}{\sqrt{\beta_T}}\right)^2 > c_5 \beta_T d \log T \cdot 4^\ell \cdot \frac{4^{-\ell}}{\beta_T} = c_5 d \log T$$

With $c_5 = 2$, this gives:

$$\sum_{t=1}^{T} \|a_t\|_{V_{t-1}^{-1}}^2 > 2d \log T$$

which contradicts Lemma 3. Hence, the assumption is false and the result holds. $\square$

**Lemma 11.** *There exists a constant $c_6 = 4/3$ such that:*

$$T < c_6 d \cdot 4^L \quad and \quad T > 0.25 d \cdot 4^L$$

*Proof.* From the definition of $L$:

$$T \leq \sum_{\ell=1}^{L} d \cdot 4^\ell \leq c_6 d \cdot 4^L$$

The lower-bound is obvious:

$$T > \sum_{\ell=1}^{L-1} d \cdot 4^\ell \geq 0.25 d \cdot 4^L$$

$\square$

**Lemma 12.** *There exists a constant $c_7$ such that:*

$$\hat{V} \preceq c_7 \beta_T d \log T \cdot V^*$$

*Proof.* In the proof below, we apply Lemma 1 to the (possibly approximate) G-optimal design guaranteed. The $\varepsilon$-approximation only affects the constants by a factor $(1 \pm \varepsilon)$, which we absorb into the hidden constants.

Define $V_{P_\ell} = \sum_{t \in P_\ell} a_t a_t^\top$, then $V = I + \sum_{\ell=1}^{L} V_{P_\ell}$.

For any $\ell \in [1, L]$, since $\{a_t \mid t \in P_\ell\} \subseteq A_l$, by Lemma 1:

$$\frac{V_{P_\ell}}{|P_\ell|} \preceq \frac{dV_\ell^*}{d \cdot 4^\ell} = \frac{V_\ell^*}{4^\ell}$$

Thus:

$$V_{P_\ell} \preceq \frac{|P_\ell| V_\ell^*}{4^\ell}$$

For $\ell \in [1, L-1]$, by Lemma 10:

$$V_{P_\ell} \preceq \frac{c_5 \beta_T d \log T \cdot 4^\ell V_\ell^*}{4^\ell} = c_5 \beta_T d \log T \cdot V_\ell^*$$

For $\ell = L$, by Lemma 11:

$$V_{P_L} \preceq \frac{TV_L^*}{4^L} \preceq \frac{c_6 d \cdot 4^L V_L^*}{4^L} \preceq c_6 d \cdot V_L^*$$

Let $c_7 = \max\left\{c_5, \frac{c_6}{\beta_T \log T}\right\}$. Then for all $\ell \in [1, L]$:

$$V_{P_\ell} \preceq c_7 \beta_T d \log T \cdot V_\ell^*$$

Summing over $\ell$:

$$\hat{V} = I + \sum_{\ell=1}^{L} V_{P_\ell} \preceq I + c_7 \beta_T d \log T \cdot \sum_{\ell=1}^{L} V_\ell^* \preceq c_7 \beta_T d \log T \cdot V^*$$

$\square$

**Lemma 13.** *For any symmetric positive definite matrix $V \in \mathbb{R}^{d \times d}$ and nonzero vector $a \in \mathbb{R}^d$:*

$$V \succeq \frac{a a^\top}{a^\top V^{-1} a}$$

*Proof.* For any $x \in \mathbb{R}^d$:

$$x^\top V x - x^\top \frac{a a^\top}{a^\top V^{-1} a} x = x^\top V x - \frac{(a^\top x)^2}{a^\top V^{-1} a}$$

By the Cauchy-Schwarz inequality for positive definite matrices:

$$(a^\top x)^2 \le (x^\top V x)(a^\top V^{-1} a)$$

Thus:

$$x^\top V x - \frac{(a^\top x)^2}{a^\top V^{-1} a} \ge 0$$

which completes the proof. $\square$

**Lemma 14.** *There exists a constant $c_8$ such that for all $\ell \in [1, L]$ and $a \in P_\ell$:*

$$a^\top \hat{V}^{-1} a \le c_8 d \log^2 T \cdot 4^{-\ell}$$

*Proof.* For $\ell = 1$:
$$a^\top \hat{V}^{-1} a \le a^\top I^{-1} a = a^\top a \le 1 < c_8 d \log^2 T \cdot 4^{-1}$$

For $\ell \in [2, L]$ and $a \in P_\ell$:

**Case 1:** If $\Delta_a > \frac{2 C_R d \log T}{\sqrt{T}}$, by Lemma 8:

$$\|a\|_{\hat{V}^{-1}} \le c_4 \Delta_a < c_4 \cdot 2^{-\ell+2}$$

Thus:
$$a^\top \hat{V}^{-1} a \le 16 c_4^2 \cdot 4^{-\ell}$$

**Case 2:** If $\Delta_a \le \frac{2 C_R d \log T}{\sqrt{T}}$, by Lemma 7:

$$\|a\|_{\hat{V}^{-1}} \le \frac{c_2 d \log T}{\sqrt{T}}$$

By Lemma 11, $T > 4 d \log k \cdot 4^L$, so:

$$a^\top \hat{V}^{-1} a \le \frac{c_2^2 d^2 \log^2 T}{T} < \frac{c_2^2 d^2 \log^2 T}{4 d \log k \cdot 4^L} \le c_2^2 d \log^2 T \cdot 4^{-\ell}$$

Taking $c_8 = \max\left\{c_2^2, \frac{16 c_4^2}{d \log^2 T}\right\}$ completes the proof. $\square$

**Lemma 15.** *There exists a constant $c_9 = 2c_8$ such that:*

$$\hat{V} \succeq \frac{1}{c_9 d^4 L \log^2 T} V^*$$

*Proof.* For any $\ell \in [1, L]$ and $a \in \text{Supp}(\pi_l)$, by Lemma 13:

$$\hat{V} \succeq \frac{aa^\top}{a^\top \hat{V}^{-1} a}$$

By Lemma 14:

$$a^\top \hat{V}^{-1} a \leq c_8 d \log^2 T \cdot 4^{-\ell}$$

Thus:

$$aa^\top \preceq c_8 d \log^2 T \cdot 4^{-\ell} \cdot \hat{V}$$

The noiseless covariance matrix is:

$$V_\ell^* = \sum_{a \in \text{Supp}(\pi_l)} d \cdot 4^\ell \cdot aa^\top$$

Since $|\text{Supp}(\pi_l)| = O(d)$ by Carathéodory's theorem:

$$V_\ell^* \preceq O(d) \cdot d \cdot 4^\ell \cdot c_8 d \log^2 T \cdot 4^{-\ell} \hat{V} \preceq O(d^3 \log^2 T \cdot) \hat{V}$$

Summing over $\ell$:

$$V^* = \sum_{\ell=1}^{L} V_\ell^* \preceq O(d^3 L \log^2 T) \cdot V$$

$\square$

### E.1 PROOF OF THEOREM 5

*Proof.* **Feasibility.** By Lemma 6, w.p. $1 - \delta_{\text{UCB}}$, $a_t^\top \theta^* \in [a_t^\top \hat{\theta}_{t-1} \pm \sqrt{\beta_t} \|a_t\|_{V_{t-1}^{-1}}]$. Since LinUCB implies $a_t^\top \hat{\theta}_{t-1} + \sqrt{\beta_t} \|a_t\|_{V_{t-1}^{-1}} \geq \mu^* \geq \mu - \kappa$, we have $a_t^\top \theta^* \geq \mu - \kappa - 2\sqrt{\beta_t} \|a_t\|_{V_{t-1}^{-1}}$. Setting $C = O(\sqrt{\beta_T})$ ensures $\theta^*$ is feasible.

**Decomposition.** Let $\hat{\theta}$ and $\hat{\theta}'$ be feasible solutions with slack $\kappa$ and $0$, respectively. Assume bounded parameters $\|\hat{\theta}\|_2, \|\hat{\theta}'\|_2 \leq \sqrt{d}$. Define $\Delta := \hat{\theta} - \theta^*$, $\Delta' := \hat{\theta}' - \theta^*$, and $\delta := \hat{\theta} - \hat{\theta}'$. Then $|a^\top \Delta| \leq |a^\top \Delta'| + |a^\top \delta|$.

**Statistical Error ($|a^\top \Delta'|$).** With $\kappa = 0$, $\langle \Delta', a_t \rangle \leq 2\sqrt{\beta_T} \|a_t\|_{V_{t-1}^{-1}}$. Using Lemma 3 and $\|\Delta'\|_2^2 \leq 4d$:

$$\|\Delta'\|_{\hat{V}}^2 = \sum_{t=1}^{T} \langle \Delta', a_t \rangle^2 + \|\Delta'\|_2^2 \leq 8\beta_T d \log T + 4d.$$

Thus, $|a^\top \Delta'| \leq \|\Delta'\|_{\hat{V}} \|a\|_{\hat{V}^{-1}} = O\left(\sqrt{d^2 \log^2 T \log(1/\delta_{\text{UCB}})}\right) \|a\|_{\hat{V}^{-1}}$.

**Perturbation Error ($|a^\top \delta|$).** Constraints imply $|\langle \delta, a_t \rangle| \leq \kappa$. Let $\mathcal{S} = \text{span}(\{a_t\})$. Decompose $a = a_\parallel + a_\perp$ where $a_\parallel \in \mathcal{S}, a_\perp \perp \mathcal{S}$.

1. *In-span ($a_\parallel$):* Using G-optimal design properties on $\mathcal{S}$, $\|a_\parallel\|_{V(\pi)^{-1}} \leq \sqrt{d}$. Thus:

$$|\langle \delta, a_\parallel \rangle| \leq \|\delta\|_{V(\pi)} \|a_\parallel\|_{V(\pi)^{-1}} \leq \sqrt{\sum \pi_i \kappa^2} \sqrt{d} = \sqrt{d}\kappa.$$

2. *Out-of-span ($a_\perp$):* Since $\hat{V} a_\perp = a_\perp$, $\|a_\perp\|_{\hat{V}^{-1}} = \|a_\perp\|_2$. With $\|\delta\|_2 \leq 2\sqrt{d}$:

$$|\langle \delta, a_\perp \rangle| \leq \|\delta\|_2 \|a_\perp\|_2 \leq 2\sqrt{d} \|a\|_{\hat{V}^{-1}}.$$

**Conclusion.** Absorbing the $a_\perp$ term into the statistical error constant:

$$|a^\top \Delta| \leq O\left(\sqrt{d^2 \log^2 T \log(1/\delta_{\mathsf{UCB}})}\right) \|a\|_{\hat{V}^{-1}} + \sqrt{d}\kappa.$$

$\square$

### E.2 PROOF OF THEOREM 6

Set $\nu = 1 - \dfrac{1}{1+c}$ in Lemma 6

*Proof.* By Lemma 12:

$$\overline{V} \preceq (1 - \delta_{\mathsf{UCB}})c_7 \beta_T d \log T \cdot V^* + \delta_{\mathsf{UCB}}(T+1)I$$

By Lemma 15, with probability at least $1 - \delta_{\mathsf{UCB}}$:

$$\hat{V} \succeq \frac{1}{c_9 d^3 L \log^2 T} V^*$$

Combining these inequalities and noting that $V^* \succeq I$ and $L = O(\log T)$:

$$\overline{V} \preceq O\left(d^5 \log^5 T \log\left(\frac{1}{\delta_{\mathsf{UCB}}}\right)\right) \hat{V}$$

$\square$

### E.3 PROOF OF THEOREM 7

*Proof.* **Feasibility.** From Theorem 4 and Theorem 5, the true parameter $\theta^*$ satisfies the individual bounds for Phased Elimination and LinUCB. Thus, with $C = O(\sqrt{\beta_T/d} + \log(L/\delta_{\mathsf{PE}}))$, $\theta^*$ satisfies the unified constraint:

$$\mu - C \cdot \max\left(\sqrt{d}\|a_t\|_{V_{t-1}^{-1}}, \sqrt{\frac{d^2}{t}}\right) - \kappa \leq \hat{\theta}^\top a_t \leq \mu$$

for all $t \in [1, T]$, ensuring the LP is feasible.

**Decomposition.** Let $\hat{\theta}$ and $\hat{\theta}'$ be feasible solutions with slack $\kappa$ and $0$, respectively, satisfying $\|\hat{\theta}\|_2, \|\hat{\theta}'\|_2 \leq \sqrt{d}$. Define $\Delta := \hat{\theta} - \theta^*$, $\Delta' := \hat{\theta}' - \theta^*$, and $\delta := \hat{\theta} - \hat{\theta}'$. We bound the error as $|a^\top \Delta| \leq |a^\top \Delta'| + |a^\top \delta|$.

**Statistical Error** ($|a^\top \Delta'|$). For the solution without slack ($\kappa = 0$), we have $|\langle \Delta', a_t \rangle| \leq C \max(\sqrt{d}\|a_t\|_{V_{t-1}^{-1}}, \sqrt{d^2/t})$. Squaring and summing over $t$, and using $\max(x, y)^2 \leq x^2 + y^2$:

$$\sum_{t=1}^T \langle \Delta', a_t \rangle^2 \leq C^2 \sum_{t=1}^T \left(d\|a_t\|_{V_{t-1}^{-1}}^2 + \frac{d^2}{t}\right)$$
$$\leq C^2 \left(2d^2 \log T + d^2(1 + \log T)\right)$$
$$= O(C^2 d^2 \log T).$$

Adding the parameter bound $\|\Delta'\|_2^2 \leq 4d$, we have $\|\Delta'\|_{\hat{V}} = O(Cd\sqrt{\log T})$. Thus:

$$|a^\top \Delta'| \leq \|\Delta'\|_{\hat{V}} \|a\|_{\hat{V}^{-1}} = O(Cd\sqrt{\log T})\|a\|_{\hat{V}^{-1}}.$$

**Perturbation Error** ($|a^\top \delta|$). The analysis of the $\kappa$-term is identical to the argument used in the proof of Theorem 2 and Theorem 5. Hence, by the same reasoning we obtain

$$|\langle \delta, a \rangle| \leq \sqrt{d}\kappa + 2\sqrt{d}\|a\|_{\hat{V}^{-1}}.$$

**Conclusion.** Combining the terms yields:

$$|a^\top \Delta| \leq O(Cd\sqrt{\log T})\|a\|_{\hat{V}^{-1}} + \sqrt{d}\kappa.$$

Substituting the required $C$ for each case:

1. For Phased-Elimination, $C = O(\sqrt{\log(L/\delta_{\mathsf{PE}})})$, yielding the bound $O(\sqrt{d^2 \log T \log(L/\delta_{\mathsf{PE}})})\|a\|_{\hat{V}^{-1}} + \sqrt{d}\kappa$.

2. For LinUCB, $C = O(\sqrt{\beta_T/d})$, yielding the bound $O(\sqrt{d^2 \log^2 T})\|a\|_{\hat{V}^{-1}} + \sqrt{d}\kappa$.

$\square$

## F  ADDITIONAL EXPERIMENTAL DETAILS

### F.1  ALGORITHM IMPLEMENTATION DETAILS

For our experimental evaluation, we implemented the algorithms as follows:

**Algorithm 3** was implemented using the binary search version, which demonstrated superior performance over the standard version in our preliminary comparisons (see Figure 1 in the main text). This variant maintains the same theoretical guarantees while achieving faster convergence in practice.

**Algorithm 4** was implemented strictly according to the pseudocode provided in the main text, without any modifications or optimizations. This ensures a faithful evaluation of the algorithm as theoretically described.

**Baseline Algorithm** was implemented using the code provided by Guha et al. (2024) to ensure a fair comparison. We used their publicly available implementation without modifications.

All algorithms were evaluated under identical experimental conditions.

### F.2  SEMI-SYNTHETIC DATA GENERATION

Our semi-synthetic experiments follow the methodology described in Section 7 of the main text, with additional implementation details provided here.

**Datasets:** We used two real-world datasets:

- **MovieLens 25M** Lam & Herlocker (2006); Harper & Konstan (2015): Contains 25 million ratings from 162,000 users on 62,000 movies.
- **Amazon Reviews Digital Music** Hou et al. (2024): Contains reviews and ratings for digital music products from the Amazon platform.

**Data Processing:** We randomly sampled $u = 6,000$ users and $m = 4,000$ items from each dataset, along with their associated ratings. For users with multiple ratings, we averaged their ratings per item.

**Matrix Factorization:** We performed matrix factorization using Alternating Least Squares (ALS) with regularization parameter $\lambda = 0.01$ and $d = 8$ latent factors. The optimization ran for 20 iterations or until convergence (tolerance $10^{-4}$). This produced user embedding matrix $U \in \mathbb{R}^{u \times d}$ and item embedding matrix $M \in \mathbb{R}^{m \times d}$.

**Bandit Simulation:** For each trial, we randomly selected a user embedding $U_i$ as the true parameter $\theta^*$. The item embedding matrix $M$ served as the action set. We ran each algorithm for $L = 8$ phases, with rewards generated as $r_t = \langle \theta^*, a_t \rangle + \epsilon_t$, where $\epsilon_t \sim \mathcal{N}(0, 0.02)$.

**Evaluation:** We repeated this process for 100 randomly selected users and averaged the relative error $\|\hat{\theta} - \theta^*\|_2/\|\theta^*\|_2$ at the final round of the last phase. The results, presented in Figure 2g-h of the main text, consistently show the superiority of our proposed algorithms over the baseline.

