# OpenReview forum: "Inverse Linear Bandits via Linear Programs"
_ICLR.cc/2026/Conference — Submitted to ICLR 2026_

### Official Review · Reviewer_asqs · 2025-10-20

**Soundness:** 3
**Presentation:** 3
**Contribution:** 3
**Rating:** 6
**Confidence:** 2

**Summary:**

This paper studies inverse bandit problems. Instead of learning a reward function from expert demonstrations, it aims to learn a reward function from the actions in the learning process, basically the process that the actions improve. The authors argue that this action slection process can help reveal information about the reward function. The paper formulates a (general) linear program to solve this problem and provide theoretical guarantee that the method achieves information-theoretically optimal reward recovery (up to polynomial factors).

**Strengths:**

The idea of learning a reward from process instead of learning from demonstrations is interesting. The authors rigorously formulate this problem as a linear program and theoretically guarantee the information-theoretically optimal reward recovery (up to polynomial factors). In general, the paper is well written and technically solid.

**Weaknesses:**

1. The paper assumes the access to an approximation of optimal reward value.

2. The paper assumes to know the algorithm that the demonstrator is using.

**Questions:**

Can the authors discuss how to solve the two weaknesses?

---

> ### Author Response · Authors · 2025-11-21
>
> We thank the reviewer for the thoughtful and encouraging comments. Below we address the two weaknesses raised in the review.
>
> **Question:**
> The method assumes access to an approximation of the optimal reward value.
>
> **Response:**
> In Appendix B\, we provided a formal impossibility result showing that without any information about $\mu^\ast$, recovering $\theta^\ast$ is information-theoretically impossible. The approximation of the optimal reward used in our method is therefore the weakest form of side information that still enables identifiability.
>
> **Question:**
> The method assumes knowledge of whether the demonstrator uses LinUCB or Phased Elimination.
>
> **Response:**
> We thank the reviewer for the opportunity to clarify this point. Although our analysis first presents separate estimators for LinUCB and Phased Elimination, our final algorithm does not require knowing which algorithm the demonstrator used. In Section 4, we introduce a program that accepts only the action sequence as input and outputs a valid estimate $\hat{\theta}$.
>
> Regarding incorporating algorithms beyond LinUCB and Phased Elimination, we note that it would be difficult to incorporate a wide range of algorithms, e.g. any no-regret linear bandits algorithm. One example of "no-regret" algorithm is Explore-Then-Commit algorithm, which is mentioned in Section 3.2 of Guha et al.(2021).
> Although Explore-Then-Commit is a no-regret algorithm with regret $O(T^{2/3})$, its exploration phase does not provide sufficient information to infer the reward structure, and therefore reverse learning is not feasible for such algorithms.
> However, incoporating algorithms like Thompson Sampling is indeed a future direction, which was left as an open problem in our paper.
>
> We thank the reviewer again for the positive evaluation and constructive questions. We hope the clarifications above address both concerns raised.

---

> > ### Comment · Reviewer_asqs · 2025-11-25
> >
> > Thanks for your response. I have decided to maintain my positive rating. Good luck!

---

### Official Review · Reviewer_abJb · 2025-10-21

**Soundness:** 3
**Presentation:** 2
**Contribution:** 2
**Rating:** 4
**Confidence:** 4

**Summary:**

The goal of this paper is to provide provably efficient algorithms for the recovery of the unknown reward function used by a linear bandit algorithm during its execution. The paper specifically assumes to observe a single demonstration from either linUCB of phase elimination, and building upon Guha et al. it derives various provably efficient algorithms for both settings, as well as a single unifying algorithm. All the algorithms are based on the same idea that each observed action provides a linear constraint on the true reward, so all algorithms simply consist in solving an LP.

**Strengths:**

- The paper provides a strong theoretical analysis of all the proposed algorithms, along with a lower bound.
- The idea of presenting a single unifying algorithm for both LinUCB and phased elimination is nice.
- All algorithms are also computationally efficient.

**Weaknesses:**

- The main limitation of this work is the scope. It is not clear in which realistic settings we can apply the modelling assumptions of this paper, i.e., that we want to recover the *linear* reward from someone that is using exactly an algorithm between phased elimination and linUCB. Also, for what should we use this linear parameter? Moreover, this paper requires knowledge of $\mu^*$ (or, at least, of an interval containing it).
- The bounds provided in the paper are very large: e.g., $d^8$. Moreover, it is not clear why the error due to $k$ reduces with more data, and I hope authors can clarify this point.
- The presentation of the paper is quite poor. Although the paper extends the work of Guha et al., the formulation of the problem and the presentation of the results is often quite imprecise. To make an example, Theorem 1 is written so badly (as well as its proof and also the proof of Theorem 2, where $\Delta$ is undefined. I did not check the others).

See also my questions below.

**Questions:**

- lines 42-43: why optimal expert leads to high sample complexity?
- lines 49-53: I would not say that assuming the expert is learning provides a practical advantage against assuming an optimal expert. Of course this holds in case the expert is actually learning, but in case the expert is not, this modelling assumption might introduce non-neglectable misspecification error, that cannot be dropped with more samples (while it can be reduced with more samples as long as we assume the expert to behave optimally).
- I do not get Theorem 1. What is parameter $\theta'$? The fact that the lower bound holds for the maximum error also with $\theta'$ seems very weird. If $\theta'$ refers to another problem instance, please, rewrite completely Theorem 1 (and also its proof, which is written bad) to allow a reader understand it. Moreover, can you clarify the difference between your Theorem 1 and Theorem 5.1 of Guha et al.?
- I do not get how you obtain the expression in line 285, because I would expect an additional $d$. Can you please show how you upper bound $\|a\|_{\hat{V}^{-1}}\le d \|_{\overline{V}^{-1}}$ knowing the relation in Theorem 3?
- Why all the efforts for assuming $\mu\in[\mu^*,\mu^*+k]$ in the paper? This requirement does not seem to improve much the generality of the method w.r.t. assuming to know $\mu^*$ directly as in Guha et al.; indeed, the extension of your algorithms to this assumption $\mu\in[\mu^*,\mu^*+k]$ instead of directly knowing $\mu^*$ is quite trivial, except for the theoretical guarantees. About this, it seems very weird to me that, e.g., in Algorithm 3, the error due to $k$ disappears as we collect more data from the expert. I would expect it to provide a fixed independent approximation error term. Can you please clarify better this point?

typos:
- line 194: I guess when $\mu^*$ is unknown
- line 253 misses the term in $k$
- $\Delta$ is never defined neither in the paper nor in the appendix, but used a lot

If you will address all my concerns (clarify the importance of the work, clarify that improving the bounds is not trivial, improve the writing, show that the error due to $k$ indeed reduces with more data), then I will increase the score to 6.

---

> ### Author Response · Authors · 2025-11-21
>
> We sincerely thank the reviewer for the detailed feedback.
>
> **Question:**
> The bounds are extremely large; unclear why the error due to $\kappa$ decreases with more data.
>
> **Response:**
> In the revision we improved the analysis and reduced the bound from $d^8$ to $d^7$ in the analysis of LinUCB. The reason for the large polynomial dependence is structural: the final estimation error consists of the product of two error sources, and each contains $d$ factors that could be difficult to eliminate. Moreover, several key steps in the analysis rely on inequalies based on the elliptical potential lemma, i.e., $\sum_{t=1}^T \lVert A_t\rVert_{V_{t-1}^{-1}}^2 \le 2 d \log T$, which inherently introduces $d$ factors. As a result, substantially reducing the $d$-dependence is far from trivial and would require new techniques.
>
> Regarding dependence on $\kappa$, in the revised version, we refined the analysis by replacing the bound $\sqrt{T}\kappa \lVert a\rVert_{V^{-1}}$ with a sharper bounds of $ \sqrt{d}\kappa$, which gives a tight dependence on $\kappa$ and is now a fixed independent approximation error term.
>
> **Question:**
> The presentation is often imprecise
>
> **Response:**
> In the revised version:
> - Theorem 1 has been rewritten for clarity.
> - All undefined parameters and symbols are now explicitly introduced.
> - The proofs of Theorem 1 and Theorem 2 were rewritten.
>
> We hope these revisions improve readability.
>
> **Question:**
> The parameter $\theta'$ is unclear; Theorem 1 seems odd; relation to Guha et al. is not well stated.
>
> **Response:**
> The revised Theorem 1 now explicitly defines $\theta'$, and clearly states the problem instance over which the bound holds.
>
> Guha et al. (2024) establish a lower bound of the form $\lVert\theta^\ast - \hat{\theta}\rVert_2 = \max_{x:\lVert x\rVert_2 = 1} x^\top(\theta^\ast - \hat{\theta}),$
> which characterizes the estimation error in the **single worst-case direction**. Their result shows that there exists at least one direction for which any inverse estimator must incur $
> \widetilde{\Omega}\left(\sqrt{d/T}\right)$ error.
>
> In contrast, our analysis provides a **per-action** characterization of the estimation error. For every action $a$, we obtain the bound$ |a^\top(\theta^\ast - \hat{\theta})| \geq \frac{1}{2}\lVert a\rVert_{V^{-1}}$, which quantifies the approximation error for each individual action rather than only for the worst-case direction. Moreover, our lower bound is demonstrator-aware, i.e., it depends on the expected feature covarince matrix of the demonstrator.
>
> When specializing to multi-armed bandits, the lower bound in Theorem 1 is equivalent to the hardness result in Guo et al. (2021) (which works only for multi-armed bandits). Therefore, Theorem 1 could be seen as a generalization of the hardness result in Guo et al. (2021) to the linear bandits setting.
>
> **Question:**
> Why does an optimal expert lead to high sample complexity?
>
> **Response:**
> When the demonstrator’s policy is (near-)optimal, a single trajectory provides poor directional coverage, so recovering the underlying rewards typically requires multiple independent demonstrations. This stands in stark contrast to our setting, where we focus on one-shot recovery from a single demonstration. We have clarified this point in the revised version.
>
> **Question:**
> I would not say that assuming the expert is learning provides a practical advantage against assuming an optimal expert.
>
> **Response:**
> Thank you for pointing this out. In the revised version, we avoid the “practical advantage” wording and make it clear that this is a complementary regime that may not be universally more practical. Our focus is that, in this regime, a single demonstration can be information-theoretically identifiable, whereas under a stationary optimal policy one typically needs multiple demonstrations or injected randomness.
>
> **Question:**
> How do we obtain $\lVert a\rVert_{\hat{V}^{-1}} \le d \lVert a\rVert_{\overline{V}^{-1}}$ from Theorem 3?
>
> **Response:**
> Using Theorem 3, we have: $ \overline{V} \preceq O(d^2)\hat{V}$, which implies $\hat{V}^{-1} \preceq O(d^2)\overline{V}^{-1}$. Since  $ \lVert a\rVert_{\hat{V}^{-1}}=\sqrt{a^\top \hat{V}^{-1}a} $, we obtain  $\lVert a\rVert_{\hat{V}^{-1}} \le O(d)\lVert a\rVert_{\overline{V}^{-1}} $
>
> **Question:**
> Why assumption $\mu \in [\mu^\ast, \mu^\ast+\kappa]$; would expect it to provide a fixed independent error term.
>
> **Response:**
> Assuming an approximate value of $\kappa$ is a more general assumption; the case $\kappa=0$ corresponds to knowing $\mu^\ast$ exactly.
>
> Regarding dependence on $\kappa$, in the revised version, we additionally refined the analysis by replacing the bound $\sqrt{T}\kappa \lVert a\rVert_{V^{-1}}$ with a sharper bounds of $ \sqrt{d}\kappa$, which gives a tight dependence on $\kappa$ and is now a fixed independent approximation error term.
>
> **Question:**
> Typos and Missing Terms
>
> **Response:**
> All have been fixed in the revised version.

---

> > ### Comment · Reviewer_abJb · 2025-11-26
> >
> > I thank the Authors for having updated the submission by incorporating my suggestions. However, you did not address the first weakness (the scope), which is very important in my opinion. Moreover, the coarse bounds ($d^7$) and the error in the theorem that the error term due to $k$ go to zero for more data, suggest me that this paper may not be ready for publication at the current moment. Put it differently, I appreciate the idea, but, *at the current time*:
> > - it is not motivated well enough
> > - the theoretical analysis should be refined
> > - the error in a theorem unfortunately suggests that there might be others
> >
> > I will thus keep my score, being slightly biased toward reject

---

> > > ### Author Response · Authors · 2025-11-28
> > >
> > > **1. Motivation and Scope**
> > >
> > > **(1) Motivation and Realistic Settings (Why Inverse Linear Bandits?)**
> > > Our setting, recovering rewards from a learning agent [1-3], addresses critical scenarios where standard IRL (assuming a fixed optimal expert) fails. We highlight two key "realistic settings":
> > >
> > > * **Auditing and Safety:** Consider a regulator auditing an online platform (e.g., ad recommender) to verify it is not optimizing for prohibited objectives (e.g., discrimination or addiction). The platform's algorithm is constantly learning and updating. Our framework allows the auditor to infer the underlying reward function from the sequence of decisions (logs) without needing access to the platform's private reward signals.
> > >
> > > * **Preference Elicitation from Human Learning:** Humans are often not optimal but "improving" agents when interacting with a new interface. Modeling the user as a learning agent (as in Inverse Bandits) allows us to recover their true preferences more accurately than assuming they are instantly optimal.
> > >
> > > **(2) Scope of Algorithms (Why LinUCB and Phased Elimination?)**
> > > While we agree that covering "any" algorithm is ideal, it is theoretically established that inverse reward estimation is impossible for arbitrary no-regret algorithms due to lack of identifiability (Guo et al., 2021), and the demonstrator should satisfy some adaptivity constraints. Therefore, we must focus on structural assumption about the demonstrator.
> > >
> > > * **Representative Paradigms:** Phased Elimination and LinUCB are not just two random algorithms; they represent the two fundamental paradigms of efficient exploration: Elimination-based and Optimism-in-the-Face-of-Uncertainty (OFU).
> > >
> > > * **A Unified Framework:** A key contribution of our work is a unified Linear Programming approach that handles both paradigms. This is a significant technical leap over prior work (Guha et al., 2024), which was restricted to elimination-based strategies. This suggests our framework has the potential to generalize to other algorithms satisfying similar confidence bound properties.
> > >
> > > **(3) Utility of the Recovered Parameter**
> > > Recovering the linear parameter $\theta$ allows us to:
> > >
> > > * **Predict Behavior:** Predict how the agent will act in unseen contexts (new action sets), which is impossible by simply mimicking the policy.
> > >
> > > * **Transfer Learning:** Use the inferred reward to train a new, more efficient agent (e.g., with a different algorithm or constraints).
> > >
> > > [1] Alexis Jacq, Matthieu Geist, Ana Paiva, and Olivier Pietquin. Learning from a learner. ICML 2019.
> > >
> > > [2] Giorgia Ramponi, Gianluca Drappo, and Marcello Restelli. Inverse reinforcement learning from a gradient-based learner. NeurIPS 2020.
> > >
> > > [3] Ashwin Balakrishna, Brijen Thananjeyan, Jonathan Lee, Felix Li, Arsh Zahed, Joseph E. Gonzalez, Ken Goldberg. On-policy robot imitation learning from a converging supervisor. CoRL 2020.
> > >
> > > **2. Requires knowledge of $\mu^*$**
> > >
> > > In Appendix B, we provided a formal impossibility result showing that without any information about $\mu^\*$, recovering $\theta^\*$ is information-theoretically impossible. The approximation of $\mu^\*$ is therefore the weakest form of side information that still enables identifiability.
> > >
> > > **3. Clarification on the $\kappa$ Term**
> > >
> > > We believe there is a misunderstanding regarding the $\kappa$ term mentioned in your comment ("error term due to $\kappa$ go to zero for more data"). We would like to clarify that $\kappa=O(d\log T/\sqrt{T})$ in Theorem 1 is a condition required by our information-theoretic lower bound, not a claim about the performance of the demonstrators or the inverse learning algorithms.
> > >
> > > * **In the Lower Bound (Theorem 1):** We explicitly state that the lower bound holds even if the estimator is given an approximation of $\mu^\*$. Theorem 1 assumes the hypothetical inverse learner has near-perfect prior knowledge of $\mu^\*$, meaning $\kappa$ approaches zero. We show that even under this assumption, the fundamental error in reward identification still exists (the lower bound holds). Therefore, the lower bound must certainly hold for the standard, information-limited inverse setting where rewards are hidden.
> > > * **In the Upper Bound (Theorem 7·):** Conversely, our proposed etimators (Algorithms 3-6) do not assume $\kappa$ vanishes with $T$. In revised Theorem 7, $\kappa$ is a fixed input parameter, and the estimation error bound related to $\kappa$ is $\sqrt{d}\kappa$, which remains constant as $T$ changes.
> > >
> > > **4. Polynomial Bounds ($d^7$)**
> > >
> > > We acknowledge the high exponent and have already made efforts in the revision to reduce the LinUCB bound. Further substantially tightening the polynomial dependence is non-trivial. Reducing the $d$-dependence may require introducing new techniques beyond the scope of optimizing existing proofs. Our primary contribution is establishing the first framework where the estimation error matches the information-theoretic lower bound (up to polynomial factors in $d$ and $\log T$).

---

### Official Review · Reviewer_cMoR · 2025-10-27

**Soundness:** 4
**Presentation:** 4
**Contribution:** 4
**Rating:** 8
**Confidence:** 5

**Summary:**

The paper studies the inverse linear bandit problem, where the goal is to estimate the underlying reward function (i.e., the linear parameter vector) from a sequence of actions taken by a demonstrator. Unlike traditional inverse reinforcement learning (IRL) settings, where the demonstrator is assumed to be optimal, this work considers demonstrators that follow no-regret learning algorithms. Specifically, the authors analyze two well-known stochastic linear bandit algorithms: LinUCB and Phased Elimination. They provide consistent estimators for the true reward parameters under both algorithms.
Building on this, the paper introduces a unified reward estimation approach that does not require prior knowledge of which demonstrator algorithm (LinUCB or Phased Elimination) generated the actions. Finally, the authors empirically evaluate their method against the benchmark from Guha et al., demonstrating improved reward estimation performance on both simulated and semi-synthetic datasets.

**Strengths:**

The paper presents a strong and original contribution for inverse linear bandits, offering new theoretical and algorithmic insights into reward estimation from no-regret demonstrators. The proposed linear programs for reward estimation under both Phased Elimination and LinUCB are novel and address several open problems in Guha et al:

1. General action sets: The paper removes prior restrictions on the density and geometry of the action set, demonstrating that consistent reward estimation is achievable under general conditions.

2. Inverse estimation for LinUCB: The construction of an estimator for LinUCB is technically sophisticated requiring a detailed round-by-round analysis rather unlike the phase by phase LP for Phased Elimination.

3. Unified estimator: The authors further propose a unified reward estimator that works across both demonstrators and without needing access to internal hyperparameters.

**Weaknesses:**

The paper is technically solid and very clearly written, with no methodological flaws. I raise a few minor points for completeness and clarification, in the Questions section.

**Questions:**

Q1. Could the authors elaborate on the binary search version of Algorithms 3 and 4?

Q2. What are the main challenges in extending the approach to Thompson Sampling? Is the difficulty primarily due to the stochastic nature of action selection (making it hard to define a deterministic LP constraint per timestep), whereas LinUCB and Phased elimination are deterministic given the current mean and confidence estimate?

---

> ### Author Response · Authors · 2025-11-21
>
> Thank you very much for your positive and encouraging evaluation of our work. We are grateful for your thoughtful questions and address them below.
>
> **Question:**
> Could the authors elaborate on the binary search version of Algorithms 3 and 4?
>
> **Response:**
> Thank you for the suggestion. In the revised version, we have added a clarification paragraph describing the binary search implementation in the Analysis of Algorithm 4.
>
> **Question:**
> What are the main challenges in extending the approach to Thompson Sampling?
>
> **Response:**
> The main challenge is that Thompson Sampling (TS) does not produce deterministic confidence intervals or deterministic action-selection rules. Our LP-based estimator relies on deriving explicit linear inequality constraints from confidence sets—something that is natural for LinUCB and Phased Elimination but difficult for TS. Under TS, the action-selection rule cannot be expressed as a stable, deterministic constraint in our LP formulation. We view adapting our method to TS as an important open question for future work.

---

> > ### Comment · Reviewer_cMoR · 2025-11-28
> >
> > Thank you for the clarifications, I recommend that the paper be accepted

---

### Official Review · Reviewer_j7mb · 2025-10-27

**Soundness:** 4
**Presentation:** 3
**Contribution:** 2
**Rating:** 8
**Confidence:** 3

**Summary:**

The paper deals with the inverse linear bandits problem. In this setting, the goal is to estimate the unknown reward function from a single action trajectory coming from a linear bandit algorithms, like Phased elimination or LinUCB. The paper extends previous works by providing a unified estimator for both Phased elimination and LinUCB that is based on solving a linear program with appropriate constraints. The paper first provides a version of the approach for phased elimination and LinUCB separately, together with their analysis, and finally combine them both in a unified linear program. The estimation error of the algorithms is compared to a theoretical lower bound, previously derived in the paper. The paper includes a brief empirical evaluation of the approach against prior methods.

**Strengths:**

The paper proposes an approach for inverse reward estimation in linear bandits that seems to advance over prior works on various aspect:
- It provides a unified estimator for phased elimination and LinUCB, which is quite critical in a setting where it is not obvious to assume that the learning algorithm is fully known;
- The estimator only needs (approximate) knowledge of the maximum reward and no access to hyperparameters of the algorithm.

Other strengths include:
- The theoretical analysis looks rigorous and clear (although the proofs were not checked for this review), including a lower bound that helps to weigh the factors appearing in the estimation error.
- The empirical analysis gives at least some empirical support to a theoretically grounded approach.
- The paper is well written and easy to follow.

**Weaknesses:**

I do not see any clear weakness for this paper, which looks like a relevant and sound research effort. A few minor weaknesses are:
- While the paper improves over prior works, the contribution is mostly incremental at an higher level of abstraction;
- The motivation for the problem setting does not appear to be very strong, although there are previous publications tackling a similar problem;
- The unified approach still looks like a combination of the specialized algorithms, rather than a general procedure that can cover many other algorithms with similar premises.

**Questions:**

While my evaluation of the paper is positive, I report a few questions the authors may address in their response. It is worth mentioning that none of the points below will have a significant impact on my evaluation.

- Motivation: Estimating the reward from a trajectory of a learning algorithm is motivated by the fact that the algorithms are already deployed in real-world systems, so data can be collected easily. However, if such algorithms are deployed, it is natural to believe that they are optimizing a known reward function, which makes the benefit of solving the inverse problem less clear. Instead, if a human is collecting data, it is highly unlikely they are following an algorithm like LinUCB or Phased elimination. Some applications that come to mind are the following. Perhaps a reward is known by the company deploying the algorithm, but not by the one solving the inverse estimation problem, which may be a competitor or a player in another market. Perhaps the reward the algorithm is maximizing is coming from a very complex and unknown function, so that the goal becomes to distill the reward in a simpler model. If that is the case, it would be nice to consider the misspecification in the analysis and a setting where both the actions and the rewards realizations are available.

- Lower bound: Can the authors clarify how their lower bounds relate to previous works, especially the alternative lower bound in Sec. 5 of Guha et al.?

- Can the authors discuss which factors of the estimation errors of their algorithm they believe are unavoidable (e.g., the term $O(\kappa)$ seems to come directly from the optimal reward assumption), may be overcome with a more sophisticated analysis/algorithm?

- Do the authors believe that the unification can be pushed even further to more linear bandits algorithms? TS sampling is mentioned in the conclusion. What about any "no-regret algorithm"?

- The performance of the algorithms is evaluated in term of estimation error on the worst-case action. However, an estimation error in some actions (e.g., very suboptimal actions) may be more acceptable than an estimation error in some other (e.g., close to optimal actions). I am wondering whether the evaluation metric for the estimator can be refined in this sense. One idea that comes to mind is to provide an estimate that minimizes the probability of producing an action sequence that is different from the one given by the ground truth rewards, especially for deterministic algorithms...

---

> ### Author Response · Authors · 2025-11-21
>
> Thank you for your thoughtful feedback and constructive comments. We appreciate the time and effort you have put into reviewing our paper. Below, we address each of your points.
>
> **Question:**
> Estimating the reward from a trajectory of a learning algorithm is motivated by the fact that the algorithms are already deployed in real-world...
>
> **Response:**
> Thank you for the thoughtful remark. We agree that when the deploying party already knows the reward, the inverse problem may be unnecessary. Our results target a complementary situation where logs are available but the reward is not accessible to the analyst (e.g., audits/third-party evaluation) or where one wishes to distill a simple linear proxy from a complex internal objective. We have adjusted the introduction to frame this as an observational regime choice (not a universal advantage) and to note that our theory focuses on algorithmic demonstrators (LinUCB or Phased elimination).
>
> **Question:**
> Can the authors clarify how their lower bounds relate to previous works
>
> **Response:**
> Guha et al. (2024) establish a lower bound of the form $\lVert\theta^\ast - \hat{\theta}\rVert_2 = \max_{x:\lVert x\rVert_2 = 1} x^\top(\theta^\ast - \hat{\theta}),$
> which characterizes the estimation error in the **single worst-case direction**. Their result shows that there exists at least one direction for which any inverse estimator must incur $
> \widetilde{\Omega}\left(\sqrt{d/T}\right)$ error.
>
> In contrast, our analysis provides a **per-action** characterization of the estimation error. For every action $a$ in the action set, we obtain the bound$ |a^\top(\theta^\ast - \hat{\theta})| \geq \frac{1}{2}\lVert a\rVert_{V^{-1}}$, which directly quantifies the approximation error for each individual action rather than only for the worst-case direction. Moreover, our lower bound is demonstrator-aware, i.e., it depends on the expected feature covarince matrix of the demonstrator.
>
> When specializing to the special case of multi-armed bandits, the lower bound in Theorem 1 is equivalent to the hardness result in Guo et al. (2021) (which works only for multi-armed bandits). Therefore, Theorem 1 could be seen as a generalization of the hardness result in Guo et al. (2021) to the linear bandits setting.
>
> **Question:**
> Do the authors believe that the unification can be pushed even further to more linear bandit algorithms? TS sampling is mentioned in the conclusion.
>
> **Response:**
> In the conclusion, we mention TS as a direction for future work. TS, involves sampling from posterior distributions, and the selection scheme in TS is not as straightforward as in LinUCB, making it difficult to construct confidence intervals in the same way. TS requires different techniques for handling uncertainty, and further work is needed to integrate it into our approach.
>
> **Question:**
> What about any "no-regret" algorithm"?
>
> **Response:**
>
> One example of "no-regret" algorithm is Explore-Then-Commit algorithm, which is mentioned in Section 3.2 of Guha et al.(2021).
> Although Explore-Then-Commit is a no-regret algorithm with regret $O(T^{2/3})$, its exploration phase does not provide sufficient information to infer the reward structure, and therefore reverse learning is not feasible for such algorithms.
>
> **Question:**
>
> Can the authors discuss which factors of the estimation errors of their algorithm they believe are unavoidable, may be overcome with a more sophisticated analysis/algorithm?
>
> **Response:**
>
> We believe a dependence on $\kappa$ is unavoidable. However, as detailed in our revision, we have successfully tightened this term. We replaced the original bound $\sqrt{T}\kappa \lVert a\rVert_{V^{-1}}$ with the sharper $\sqrt{d}\kappa$ which gives a tight dependence on $\kappa$ and is now a fixed independent approximation error term.
>
> Regarding the polynomial factors in $d$, we have improved our analysis in the revision to reduce the dependence to $d^7$.However, several key steps in the analysis rely on the standard self-normalized inequality $\sum_{t=1}^T \lVert A_t\rVert_{V_{t-1}^{-1}}^2 \le 2 d \log T$, which inherently introduces $d$ factors. As a result, substantially reducing the $d$-dependence is far from trivial and would require new techniques.
>
> **Question:**
> The performance of the algorithms is evaluated in terms of estimation error on the worst-case action. However, an estimation error in some actions may be more acceptable than an estimation error in some other.
>
> **Response:**
> Thank you for this insightful suggestion. We understand that in some cases, estimation errors in suboptimal actions may be more tolerable than those in near-optimal actions. However, we chose to evaluate the performance based on the worst-case action in our experiments for consistency with the approach used in Guha et al. (2024), where the same evaluation metric was employed. This allows for a fair comparison between our results and theirs.

---

> > ### Comment · Reviewer_j7mb · 2025-11-27
> >
> > Dear Authors,
> >
> > Thank you for the detailed replies. I agree with R. abJb that the scope of the work is narrow. I still see it as a significant advancement over prior work. I will recommend acceptance.
> >
> > Best regards,
> >
> > Reviewer j7mb

---

### Author Response · Authors · 2025-11-21
**Revision**

We thank the reviewers for recognizing our contribution and for their thoughtful and detailed feedback. We have updated a revised version of the paper, with changes highlighted in red. Key changes include the followings.

* Wording changes in the introduction part regarding problem settings and motivations.

* **Tighter Bounds for LinUCB:** The analysis for LinUCB has been improved to reduce the dependence on $d$, decreasing the bound from $d^8$ to $d^7$.

* **Refined $\kappa$ Error Term:** The original bound $\sqrt{T}\kappa \lVert a\rVert_{V^{-1}}$ has been replaced with a tighter bound $\sqrt{d}\kappa$ for both phased elimination and LinUCB. This modification ensures the $\kappa$ term is a fixed error rather than one growing with $\sqrt{T}$.

* **Theorem 1 Rewritten:** Theorem 1 has been completely rewritten to improve clarity.

* **Proof Revisions:** The proofs for both Theorem 1 and Theorem 2 have been rewritten and carefully checked for correctness.

* **Binary Search Detail:** A paragraph has been added to the "Analysis of Algorithm 4" to describe the implementation of binary search.
* **Parameter Definitions:** All previously undefined parameters and symbols are now introduced.

* **Typo Corrections:** Various typos have been corrected.

---

### Meta-Review · Area_Chair_ryGx · 2026-01-02

**Summary:**

Most reviewers are in favor of this submission due to its theoretical advancement and results. However, two reviewers confirm scope and motivation remain two concerns of this paper. First, while there are previous publications solving the same problem, this work is solely motivated by limitations of Guha et al. (2024). While IRL is well motivated, why modeling the unknown reward as a linear function is not very clear. Second, if a human is collecting data, it is highly unlikely they are following an algorithm like LinUCB or Phased elimination. To address this concern, authors chose to shift the focus on algorithmic demonstrators, but in this case the scope of this work gets narrow. Finally, by writing "our setting addresses critical scenarios where standard IRL (assuming a fixed optimal expert) fails", the authors claim to work for the setting where the underlying algorithm is constantly learning and updating, which is actually different from the problem formulation in Section 3 where the sequence of actions must be produced by a single run of either Phased Elimination or LinUCB, so the motivation remains a concern. Given these unresolved concerns, I recommend Rejection.

**Reviewer Concerns:**

Reviewers raised concerns on large terms in theoretical results, presentation, scope, and motivation. Most concerns have been addressed by authors' rebuttal, but I believe scope and motivation remain two big concerns.

**Reviewer Scores:**

Reviewers j7mb, cMoR, asqs have confirmed their final ratings by reading authors' rebuttal.

Reviewer abJb did read authors' rebuttal but found that it didn't address the motivation issue, and then Reviewer abJb didn't reply to authors' second response. After carefully reading the communication between authors and Reviewer abJb, it seems that the motivation is still a big problem, so I think Reviewer abJb would keep a rating at 4.

---

### Decision · Program_Chairs · 2026-01-26

Reject